# The E3 ligase Thin controls homeostatic plasticity through neurotransmitter release repression

Martin Baccino-Calace[1,2], Katharina Schmidt[1], Martin Müller[1,2,3]*

[1]Department of Molecular Life Sciences, University of Zurich, Zurich, Switzerland; [2]Zurich Ph.D. Program in Molecular Life Sciences, Zurich, Switzerland; [3]Neuroscience Center Zurich, University of Zurich/ETH Zurich, Zurich, Switzerland

**Abstract** Synaptic proteins and synaptic transmission are under homeostatic control, but the relationship between these two processes remains enigmatic. Here, we systematically investigated the role of E3 ubiquitin ligases, key regulators of protein degradation-mediated proteostasis, in presynaptic homeostatic plasticity (PHP). An electrophysiology-based genetic screen of 157 E3 ligase-encoding genes at the *Drosophila* neuromuscular junction identified *thin*, an ortholog of human *tripartite motif-containing 32* (*TRIM32*), a gene implicated in several neurological disorders, including autism spectrum disorder and schizophrenia. We demonstrate that *thin* functions presynaptically during rapid and sustained PHP. Presynaptic *thin* negatively regulates neurotransmitter release under baseline conditions by limiting the number of release-ready vesicles, largely independent of gross morphological defects. We provide genetic evidence that *thin* controls release through *dysbindin*, a schizophrenia-susceptibility gene required for PHP. Thin and Dysbindin localize in proximity within presynaptic boutons, and Thin degrades Dysbindin in vitro. Thus, the E3 ligase Thin links protein degradation-dependent proteostasis of Dysbindin to homeostatic regulation of neurotransmitter release.

## Editor's evaluation

The paper focuses on presynaptic homeostatic plasticity (PHP) at the glutamatergic larval *Drosophila* neuromuscular synapse. In this facet of synaptic plasticity, the presynapse increases neurotransmitter release to compensate for diminished postsynaptic sensitivity. To study functional pathways and identify new molecular components of PHP, the authors carried out an electrophysiology-based genetic screen of E3 ubiquitin ligases – key regulators of protein function and degradation pathway and this screen, which forms the backbone of the paper, generated an extensive dataset encompassing 180 genotypes. In follow-up studies, the authors find that the E3 ligase Thin suppresses glutamate release, likely by targeting and downregulating Dysbindin, a transmitter-release-promoting presynaptic protein and based on the experimental data, a model is put forward according to which PHP arises by relieving Dysbindin of Thin-dependent ubiquitination and degradation. This is a strong paper that adds a highly interesting feature to the understanding of the molecular mechanisms that control synaptic strength.

*For correspondence:
Martin.Mueller@mls.uzh.ch

## Introduction

Nervous system function is remarkably robust despite continuous turnover of the proteins determining neural function. Work in nervous systems of various species has established that evolutionarily conserved homeostatic signaling systems maintain neural activity within adaptive ranges (*Marder*

*and Goaillard, 2006*; *Turrigiano, 2008*; *Delvendahl and Müller, 2019b*). Chemical synapses evolved mechanisms that compensate for neural activity perturbations through homeostatic regulation of neurotransmitter release ('presynaptic homeostatic plasticity', PHP) (*Petersen et al., 1997*; *Frank et al., 2006*; *Delvendahl and Müller, 2019b*), or neurotransmitter receptors (synaptic scaling) (*Turrigiano et al., 1998*). Several studies have established links between homeostatic control of synaptic transmission and neurological disorders, such as autism spectrum disorder (*Mullins et al., 2016*), schizophrenia (*Wondolowski and Dickman, 2013*), or amyotrophic lateral sclerosis (*Perry et al., 2017*; *Orr et al., 2020*).

Synaptic proteins are continuously synthesized and degraded, resulting in half-lives ranging from hours to months (*Cohen et al., 2013*; *Fornasiero et al., 2018*). The ubiquitin–proteasome system (UPS) is a major protein degradation pathway that controls protein homeostasis, or proteostasis. E3 ubiquitin ligases confer specificity to the UPS by catalyzing the ubiquitination of specific target proteins, thereby regulating their function or targeting them for proteasomal degradation (*Zheng and Shabek, 2017*). Synaptic proteostasis, and E3 ligases in particular, have been implicated in various neurological disorders (*George et al., 2018*). However, our understanding of the role of E3 ligases in the regulation of synaptic transmission is very limited. While several E3 ligases have been linked to postsynaptic forms of synaptic plasticity (*Hegde, 2010*), only three E3 ligases, Scrapper (*Yao et al., 2007*), Highwire (*Russo et al., 2019*), and Ariadne-1 (*Ramírez et al., 2021*) have been implicated in the regulation of presynaptic function. Moreover, a systematic investigation of E3 ligase function in the context of synaptic transmission is lacking.

PHP stabilizes synaptic efficacy in response to neurotransmitter receptor perturbation at neuro-muscular junctions (NMJs) of *Drosophila melanogaster* (*Petersen et al., 1997*; *Frank et al., 2006*; *Delvendahl and Müller, 2019b*), mice (*Wang et al., 2010*), rats (*Plomp et al., 1992*), and humans (*Cull-Candy et al., 1980*). Furthermore, there is recent evidence for PHP in the mouse cerebellum (*Delvendahl et al., 2019a*). The molecular mechanisms underlying PHP are best understood at the *Drosophila* NMJ (*Delvendahl and Müller, 2019b*), because this system is amenable to electrophysiology-based genetic screens (*Dickman and Davis, 2009*; *Müller et al., 2011*; *Delvendahl and Müller, 2019b*). At this synapse, pharmacological or genetic impairment of glutamate receptor (GluR) activity triggers a retrograde signal that enhances presynaptic release, thereby precisely compensating for this perturbation (*Petersen et al., 1997*; *Frank et al., 2006*). PHP can be induced within minutes after pharmacological receptor impairment (*Frank et al., 2006*). Severing the motoneuron axons forming the *Drosophila* NMJ in close vicinity of the NMJ does not impair PHP upon pharmacological receptor impairment (*Frank et al., 2006*), indicating that the mechanisms underlying rapid PHP expression act locally at the synapse. Moreover, pharmacological inhibition of protein synthesis by cyclohexi-mide does not affect PHP after pharmacological receptor impairment at the *Drosophila* NMJ (*Frank et al., 2006*), suggesting that de novo protein synthesis is not required for PHP expression on rapid time scales. By contrast, acute or sustained disruption of the presynaptic proteasome blocks PHP (*Wentzel et al., 2018*), demonstrating that presynaptic UPS-mediated proteostasis is required for PHP. Furthermore, genetic data link UPS-mediated degradation of two proteins, Dysbindin and RIM, to PHP (*Wentzel et al., 2018*). Yet, it is currently unclear how the UPS controls PHP. Based on the critical role of E3 ligases in UPS function, we hypothesized an involvement of E3 ligases in PHP.

Here, we realized an electrophysiology-based genetic screen to systematically analyze the role of E3 ligases in neurotransmitter release regulation and PHP at the *Drosophila* NMJ. This screen discovered that the E3 ligase-encoding gene *thin*, an ortholog of human *TRIM32* (*LaBeau-DiMenna et al., 2012*; *Domsch et al., 2013*), controls neurotransmitter release and PHP. We provide evidence that *thin* regulates the number of release-ready synaptic vesicles through *dysbindin*, a gene linked to PHP in *Drosophila* and schizophrenia in humans.

## Results

### An electrophysiology-based genetic screen identifies *thin*

To systematically test the roles of E3 ligases in PHP, we first generated a list of genes predicted to encode E3 ligases in *D. melanogaster*. To this end, we browsed the *D. melanogaster* genome for known E3-ligase domains (*Du et al., 2011*; *Ketosugbo et al., 2017*). Moreover, we included homo-logs of predicted vertebrate E3 ligases (see *Figure 1—figure supplement 1*). This approach yielded

281 putative E3 ligase-encoding genes (*Figure 1A*), significantly higher than previously predicted for *D. melanogaster* (207 genes; *Du et al., 2011*). To explore the relationship between the number of E3 ligase-encoding genes and the number of protein-coding genes, we plotted the number of putative E3 ligase-encoding genes over the total protein-coding gene number of three species and compared it to the relationship between protein kinase-encoding genes and genome size (*Figure 1A*). The relatively constant ratio between the predicted number of E3 ligase-encoding genes and genome size across species (~0.02–0.03; *Figure 1A*; *Ketosugbo et al., 2017*), suggests an evolutionarily conserved stoichiometry between E3 ligases and target proteins, similar to protein kinases (*Figure 1A*). Hence, a core mechanism of the UPS – protein ubiquitination – is likely conserved in *D. melanogaster*.

After prioritizing for evolutionarily conserved genes that were shown or predicted to be expressed in the nervous system (*Figure 1—figure supplement 1*), we investigated PHP after genetic perturbation of 157 putative E3 ligase genes and 11 associated genes (180 lines, *Supplementary file 1*, *Figure 1B*). Specifically, we recorded spontaneous miniature excitatory postsynaptic potentials (mEPSPs) and action potential (AP)-evoked excitatory postsynaptic currents (EPSCs) after applying subsaturating concentrations of the GluR antagonist philanthotoxin-443 (PhTX) for 10 min (20 µM; extracellular $Ca^{2+}$ concentration, 1.5 mM). At wild-type (WT) NMJs, PhTX treatment significantly reduced mEPSP amplitude compared to untreated controls (*Figure 1C*, black and gray arrows), indicating GluR perturbation. By contrast, AP-evoked EPSC amplitudes were similar between PhTX-treated and untreated WT NMJs (*Figure 1D*, black and gray arrows). Together with a reduction in mEPSP amplitude, a similar EPSC amplitude suggests a homeostatic increase in neurotransmitter release after PhTX treatment in WT, consistent with PHP (*Frank et al., 2006*). PhTX also reduced mean mEPSP amplitudes in the 180 transgenic or mutant lines (either presynaptic/neuronal RNAi expression, $elav^{c155}$-Gal4>UAS RNAi; or mutations within the respective coding sequence, see Materials and methods) compared to untreated WT controls (*Figure 1C*). Moreover, the mean EPSC amplitude of the majority of the tested lines did not differ significantly from the mean WT EPSC amplitudes recorded at PhTX-treated NMJs (*Figure 1D*, compare white bars with black arrow). The combination of a decrease in mEPSP amplitude and largely unchanged EPSC amplitude indicates that the majority of the tested lines likely display PHP. We also identified 21 transgenic or mutant lines with significantly smaller EPSC amplitudes compared to PhTX-treated WT NMJs, and two lines with increased EPSC amplitudes (*Figure 1D, E*, red data). The lines with smaller EPSC amplitudes represent candidate transgenic or mutant lines with disrupted PHP. One of the mutant lines with significantly smaller EPSC amplitudes in the presence of PhTX was a previously described deletion of the gene *thin/abba* (*tn*, *CG15105*), henceforth called *thin* (*thin^ΔA*; *LaBeau-DiMenna et al., 2012*; *Figure 1E*, filled red data). *thin* encodes an E3 ligase with a N-terminal tripartite motif (TRIM), which contains one RING-finger domain, two zinc-finger domains (B1 box and B2 box), and its associated coiled-coil region, followed by a disordered region and C-terminal NHL repeats (*Figure 1—figure supplement 2*). Based on this domain composition, *thin* likely represents the *Drosophila* ortholog of human *TRIM32* (*Figure 1—figure supplement 2*), consistent with earlier work (*LaBeau-DiMenna et al., 2012*). *thin* was selected for further analysis.

## Presynaptic *thin* promotes PHP

In the genetic screen, we compared synaptic transmission between a given genotype and WT controls in the presence of PhTX (*Figure 1C–E*). Hence, the small EPSC amplitude of *thin^ΔA* mutants seen after PhTX application could be either due to impaired PHP, or a defect in baseline synaptic transmission. To distinguish between these possibilities, we next quantified synaptic transmission in the absence and presence of PhTX in *thin^ΔA* mutants (*Figure 2*). Similar to WT controls, PhTX application significantly reduced mEPSC amplitude by ~40% in *thin^ΔA* mutants (*Figure 2A, B*), suggesting similar receptor impairment. At WT synapses, EPSC amplitudes were similar in the absence and presence of PhTX (*Figure 2A, C*). In combination with the decrease in mEPSC amplitude (*Figure 2B*), PhTX incubation increased quantal content (EPSC amplitude/mEPSC amplitude) in WT (*Figure 2D*), indicating homeostatic release potentiation. By contrast, PhTX treatment significantly reduced EPSC amplitudes in *thin^ΔA* mutants (*Figure 2A, C*) and did not increase quantal content (*Figure 2D*). These data show that *thin* is required for acute PHP expression.

To test if presynaptic or postsynaptic *thin* promotes PHP, and if the PHP defect is indeed caused by loss of *thin*, we assessed PHP after presynaptic or postsynaptic expression of a *thin* transgene in the *thin^ΔA* mutant background. PhTX treatment significantly reduced mEPSC amplitudes after neural/

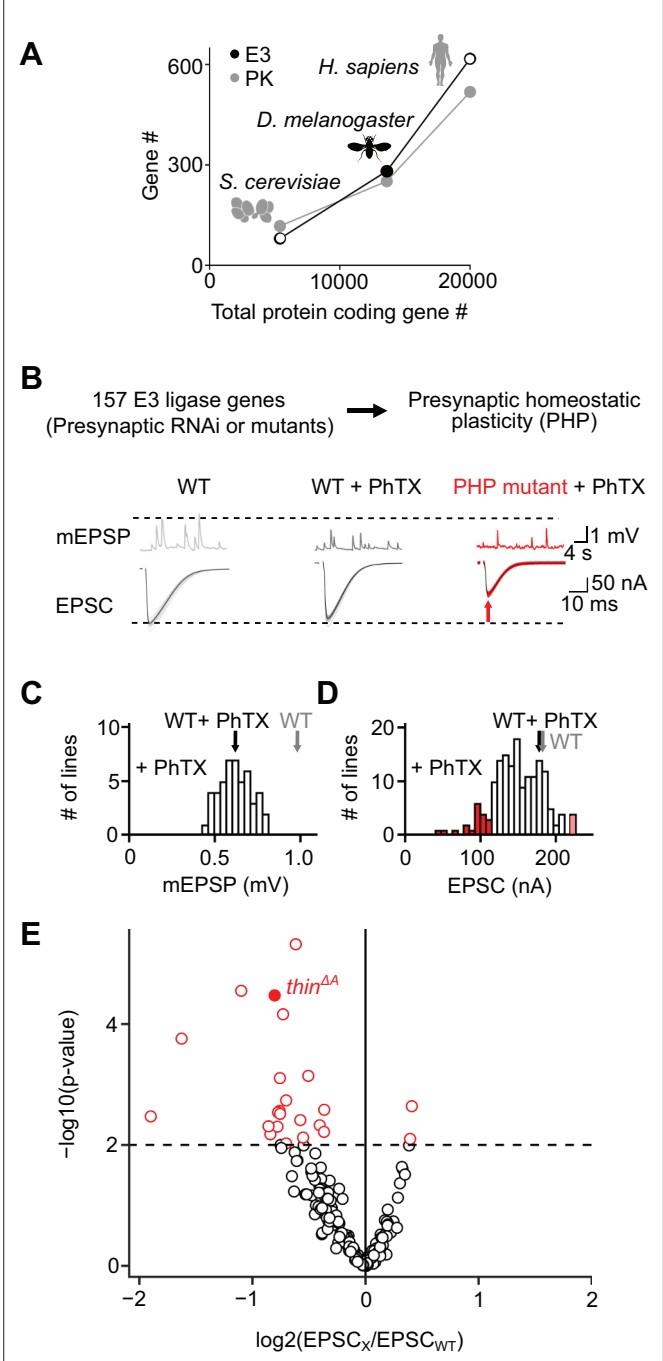

**Figure 1.** An electrophysiology-based genetic screen identifies *thin* as a synaptic homeostasis gene. (**A**) The number of putative E3 ubiquitin ligase-encoding genes (E3) and protein kinase-encoding genes (PK) as a function of total protein-coding gene number of *C. cerevisiae*, *D. melanogaster*, and *H. sapiens*. Note the similar relationship between E3 number or PK number and total protein-coding gene number across species. (**B**) *Top*: 157 E3 ligase-encoding genes and 11 associated genes (180 lines; presynaptic RNA$_i$ expression, *elav*[c-155]-Gal4>*UAS RNA$_i$*, or mutants, note that some genes were targeted by more than one line) were tested using two-electrode voltage clamp analysis at the *Drosophila* neuromuscular junction (NMJ) in the presence of the glutamate receptor (GluR) antagonist philanthotoxin-443 ('PhTX') to assess presynaptic homeostatic plasticity (PHP) (see Materials and methods). *Bottom*: Exemplary miniature excitatory postsynaptic potentials (mEPSPs) and action potential (AP)-evoked excitatory postsynaptic currents (EPSCs) recorded from wild-type (WT), WT in the presence of PhTX ('WT + PhTX'), and a PHP mutant in the presence of PhTX ('PHP mutant + PhTX'). Note the decrease in mEPSP amplitude after PhTX treatment, indicating GluR inhibition, and the similar EPSC amplitude between WT and

*Figure 1 continued on next page*

*Figure 1 continued*

WT + PhTX, suggesting PHP. Small EPSC amplitudes in the presence of PhTX (red arrow) imply a defect in PHP or baseline synaptic transmission. (**C**) Histogram of mean mEPSP amplitudes for each transgenic or mutant line (mean *n* = 4 NMJs per line, range 3–12; *N* = 180 lines) following PhTX treatment. WT averages under control conditions ('WT', *n* = 16) and in the presence of PhTX ('WT + PhTX', *n* = 16) are shown as gray and black arrows, respectively. (**D**) Histogram of mean EPSC amplitudes (as in C). The red bars indicate transgenic or mutant lines with EPSC amplitudes significantly different from WT in the presence of PhTX (black arrow). (**E**) Volcano plot of the ratio between the mean EPSC amplitude of a transgenic or mutant line and WT ('EPSC$_x$/EPSC$_{WT}$') in the presence of PhTX (p values from one-way analysis of variance [ANOVA] with Tukey's multiple comparisons). Transgenic or mutant lines with mean EPSC amplitude changes with p ≤ 0.01 (dashed line) are shown in red. A deletion in the gene *thin* (*CG15105*; *thin*$^{ΔA}$; **LaBeau-DiMenna et al., 2012**) that was selected for further analysis is shown as a filled red circle. One-way ANOVA with Tukey's multiple comparisons was performed for statistical testing (**C–E**).

The online version of this article includes the following figure supplement(s) for figure 1:

**Figure supplement 1.** Generation and prioritization of the E3 ligase-encoding gene list.

**Figure supplement 2.** Homology between Thin and TRIM family proteins.

presynaptic (*elav*$^{c155}$-*Gal4*) or postsynaptic (*24B-Gal4*) expression of *thin* (*UAS-thin*) in *thin*$^{ΔA}$ mutants (**Figure 2A, B**). After presynaptic *thin* expression in *thin*$^{ΔA}$ mutants (presynaptic rescue or 'pre. rescue'), quantal content was significantly increased upon PhTX treatment (**Figure 2D**, blue data), and EPSC amplitudes were restored toward control levels in the absence of PhTX (**Figure 2A, C**, blue data). Note that the partial rescue may be due to *thin* overexpression or defects in muscle architecture (**LaBeau-DiMenna et al., 2012**). By contrast, quantal content was similar between PhTX-treated and untreated NMJs after postsynaptic *thin* expression in the *thin*$^{ΔA}$ mutant background (postsynaptic rescue or 'post. rescue'; **Figure 2D**, green data), and PhTX application reduced EPSC amplitudes (**Figure 2A, C**, green data). Thus, presynaptic, but not postsynaptic *thin* expression enhanced quantal content after PhTX treatment in the *thin*$^{ΔA}$ mutant background (**Figure 2D**), implying a presynaptic role for *thin* in PHP.

We also noted a decrease in mEPSC amplitude in *thin*$^{ΔA}$ mutants compared to WT in the absence of PhTX (**Figure 2A, B**), which is most likely due to impaired muscle architecture in *thin*$^{ΔA}$ mutants (**LaBeau-DiMenna et al., 2012**; **Domsch et al., 2013**). Postsynaptic, but not presynaptic *thin* expression, significantly increased mEPSC amplitudes toward WT levels in the *thin*$^{ΔA}$ mutant background (**Figure 2A, B**), suggesting that postsynaptic *thin* is required for normal mEPSC amplitude levels. Furthermore, *thin*$^{ΔA}$ mutants displayed a significant increase in quantal content compared to WT under baseline conditions in the absence of PhTX (**Figure 2E**), which was rescued by presynaptic, but not postsynaptic *thin* expression (**Figure 2E**). These data are consistent with the idea that presynaptic *thin* represses release under baseline conditions (see Figure 4, Figure 6). By extension, the increased release under baseline conditions in *thin*$^{ΔA}$ mutants may partially occlude PHP in response to receptor perturbation (see Discussion).

At the *Drosophila* NMJ, genetic ablation of the GluRIIA subunit in *GluRIIA*$^{SP16}$ mutants reduces quantal size and induces sustained PHP (**Petersen et al., 1997**). To test if *thin* is required for sustained PHP expression, we generated recombinant flies carrying the *GluRIIA*$^{SP16}$ and the *thin*$^{ΔA}$ mutation ('*GluRIIA*$^{SP16}$, *thin*$^{ΔA}$'). While we observed a significant increase in quantal content in *GluRIIA*$^{SP16}$ mutants compared to wild type (**Figure 2—figure supplement 1**), indicating sustained PHP expression, there was no increase in quantal content in *GluRIIA*$^{SP16}$, *thin*$^{ΔA}$ double mutants compared to *thin*$^{ΔA}$ mutants (**Figure 2—figure supplement 1**). Hence, *thin* is also necessary for sustained PHP expression, providing independent evidence for its role in homeostatic release regulation.

## Changes in NMJ development unlikely underlie the PHP defect in *thin* mutants

The PHP defect and the release enhancement under baseline conditions after presynaptic *thin* perturbation may arise from impaired synaptic development. To test this possibility, we investigated NMJ morphology in *thin* mutants (**Figure 3**). Immunostainings with an antibody detecting neuronal membrane (anti-horseradish peroxidase, 'HRP'; **Figure 3A**; **Jan and Jan, 1982**) revealed no changes in HRP area in *thin*$^{ΔA}$ mutants or after presynaptic rescue (*thin*$^{ΔA}$; *elav*$^{c155}$-*Gal4*>*UAS* thin), and a trend toward increased HRP area after postsynaptic rescue (*thin*$^{ΔA}$; *24BGal4*>*UAS* thin) compared to WT (**Figure 3B**). Analysis of the active-zone marker Bruchpilot (anti-Bruchpilot, 'Brp'; **Kittel et al., 2006**)

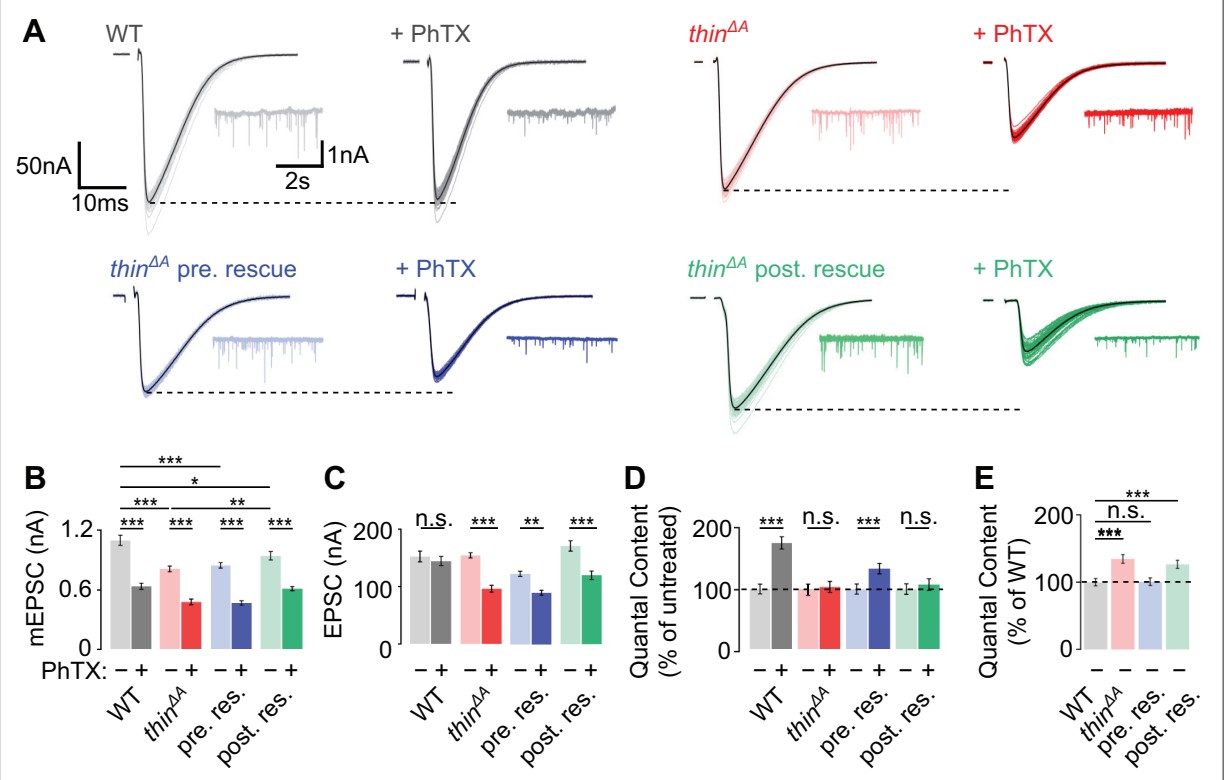

**Figure 2.** Homeostatic plasticity requires presynaptic *thin*. (**A**) Representative excitatory postsynaptic currents (EPSCs) (individual sweeps and averages are shown in light colors and black, respectively), and mEPSCs (insets) of wild-type (WT) (gray), *thin^{ΔA}* mutants (red), presynaptic *thin* expression in *thin^{ΔA}* mutants (*elav^{c155}-Gal4>UAS-thin; thin^{ΔA}*, '*thin^{ΔA}* pre. rescue', blue), and postsynaptic *thin* expression in *thin^{ΔA}* mutants (*24B-Gal4>UASthin; thin^{ΔA}*, '*thin^{ΔA}* post. rescue', green) in the absence and presence of philanthotoxin-443 (PhTX) ('+PhTX', darker colors). Stimulation artifacts were blanked for clarity. Note the decreased EPSC amplitudes at PhTX-treated *thin^{ΔA}* mutant neuromuscular junctions (NMJs) and *thin^{ΔA}* post. rescue NMJs, indicating impaired presynaptic homeostatic plasticity (PHP). Mean mEPSC amplitudes (**B**), EPSC amplitudes (**C**), quantal content after PhTX treatment normalized to the respective untreated control (**D**), in the absence ('−') and presence ('+') of PhTX, as well as baseline quantal content of the indicated genotypes in the absence ('−') of PhTX normalized to WT (**E**). Note that PhTX did not enhance quantal content in *thin^{ΔA}* mutants (**D**), indicating impaired PHP. Also note the increased quantal content under baseline conditions in *thin^{ΔA}* mutants (**E**), suggesting increased release. The PHP and baseline synaptic transmission defects are restored upon presynaptic *thin* expression in the mutant background. Mean ± standard error of the mean (SEM) (WT − PhTX: $n = 14$, WT + PhTX: $n = 13$; *thin^{ΔA}* − PhTX: $n = 18$; *thin^{ΔA}* + PhTX: $n = 21$; pre. res. − PhTX: $n = 11$; pre. res.+ PhTX: $n = 10$; post. res. − PhTX: $n = 25$; post. res.+ PhTX: $n = 24$); *$p < 0.05$; **$p < 0.01$; ***$p < 0.001$; n.s.: not significant; two-way analysis of variance (ANOVA) followed by Tukey's post hoc test (**B–D**) and one-way ANOVA with Tukey's multiple comparisons (**E**).

The online version of this article includes the following source data and figure supplement(s) for figure 2:

**Source data 1.** Related to *Figure 2*.

**Figure supplement 1.** Sustained homeostasis is impaired in *thin* mutants.

**Figure supplement 1—source data 1.** Related to *Figure 2—figure supplement 1*.

uncovered no changes in Brp puncta number per NMJ in *thin^{ΔA}* mutants or after presynaptic rescue, and a slight increase after postsynaptic rescue compared to WT (*Figure 3A, C*). Brp density (Brp puncta #/HRP area) was unchanged in *thin^{ΔA}* mutants or after postsynaptic rescue, and slightly increased after presynaptic rescue (*Figure 3D*). Finally, we observed a decrease in Brp puncta intensity in *thin^{ΔA}* mutants and upon presynaptic rescue (*Figure 3E*). In principle, these morphological alterations could be related to the PHP defect, or the release enhancement seen in *thin^{ΔA}* mutants. However, while HRP area and Brp puncta number were unchanged in *thin^{ΔA}* mutants (*Figure 3B, C*), PHP was blocked, and baseline synaptic transmission enhanced (*Figure 2*). In addition, postsynaptic *thin* expression in WT induced an increase in HRP area and Brp puncta number (*Figure 3—figure supplement 1B, C*), but neither impaired PHP nor enhanced release (*Figure 3—figure supplement 1F–K*). Furthermore, Brp intensity was decreased after presynaptic rescue (*Figure 3E*) and postsynaptic *thin* overexpression in WT (*Figure 3—figure supplement 1E*), whereas synaptic physiology was unchanged in these

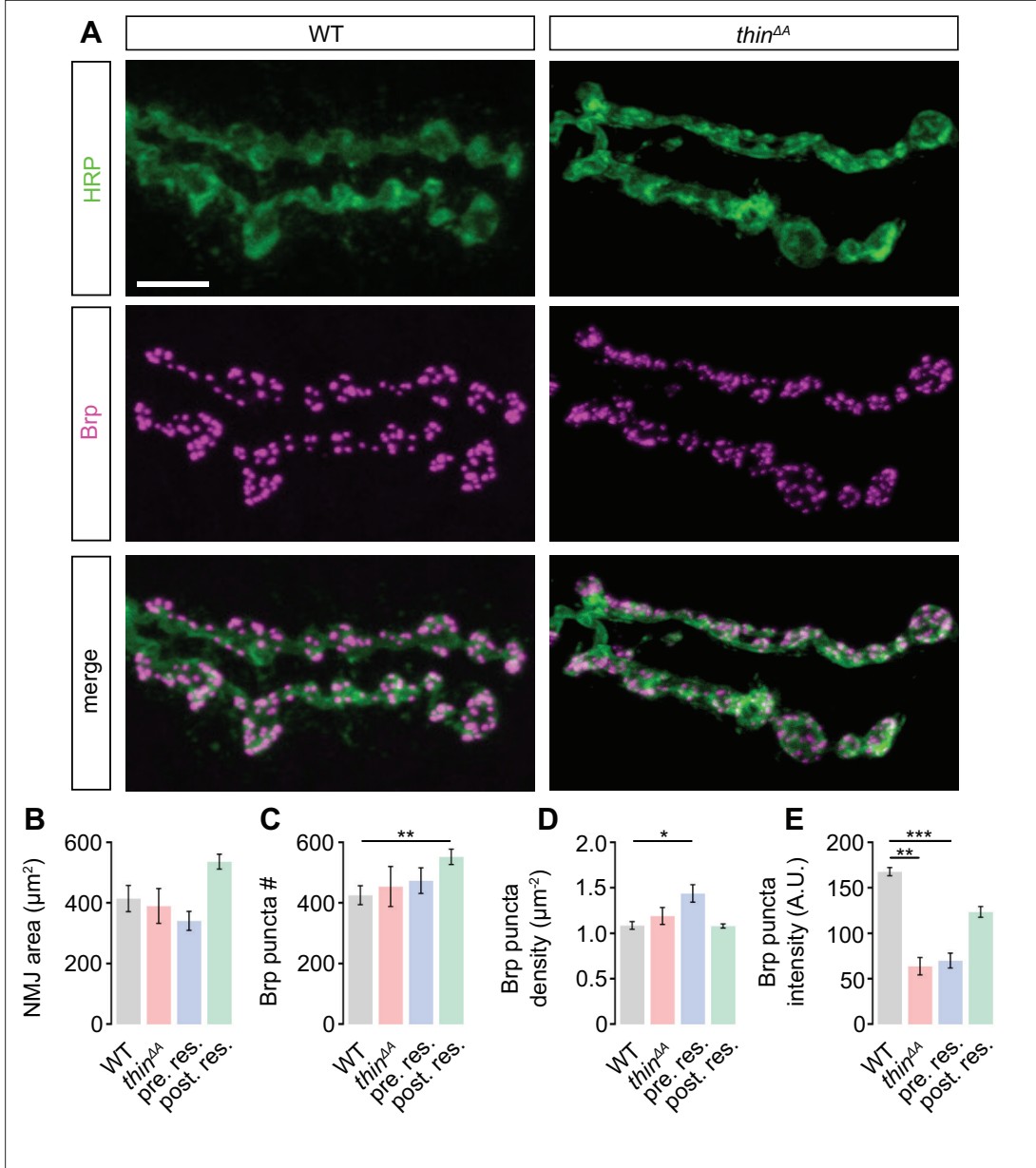

**Figure 3.** Slight alterations in neuromuscular junction (NMJ) morphology upon genetic *thin* manipulations. (**A**) Maximum intensity projection of a wild-type (WT) (left) and *thin^{ΔA}* mutant NMJ (right) (muscle 6) stained against the *Drosophila* neuronal membrane marker anti-HRP ('HRP') and the active-zone marker Bruchpilot ('Brp'); scale bar, 10 μm. Mean HRP area per NMJ 'HRP area' (**B**), Brp puncta number per NMJ 'Brp puncta #' (**C**), Brp puncta number/HRP area per NMJ 'Brp density' (**D**), and Brp puncta fluorescence intensity (**E**) of the indicated genotypes ('Postsynaptic rescue': *24B-Gal4>UAS-thin; thin^{ΔA}*; 'presynaptic rescue': *elav^{c155}-Gal4>UASthin; thin^{ΔA}*). Although changes in the recorded parameters may contribute to changes in synaptic physiology, altered NMJ morphology was separable from synaptic physiology (see Results, Discussion, *Figure 2*, *Figure 3—figure supplement 1*, *Figure 4*, *Figure 4—figure supplement 1*). Mean ± standard error of the mean (SEM); WT: *n* = 10, *thin^{ΔA}*: *n* = 8, pre. res.: *n* = 12; post. res.: *n* = 13; *p < 0.05; **p < 0.01; ***p < 0.001; n.s.: not significant; Student's *t*-test.

The online version of this article includes the following source data and figure supplement(s) for figure 3:

**Source data 1.** Related to *Figure 3*.

**Figure supplement 1.** Postsynaptic *thin* expression does not affect presynaptic homeostatic plasticity (PHP) or baseline synaptic transmission.

**Figure supplement 1—source data 1.** Related to *Figure 3—figure supplement 1*.

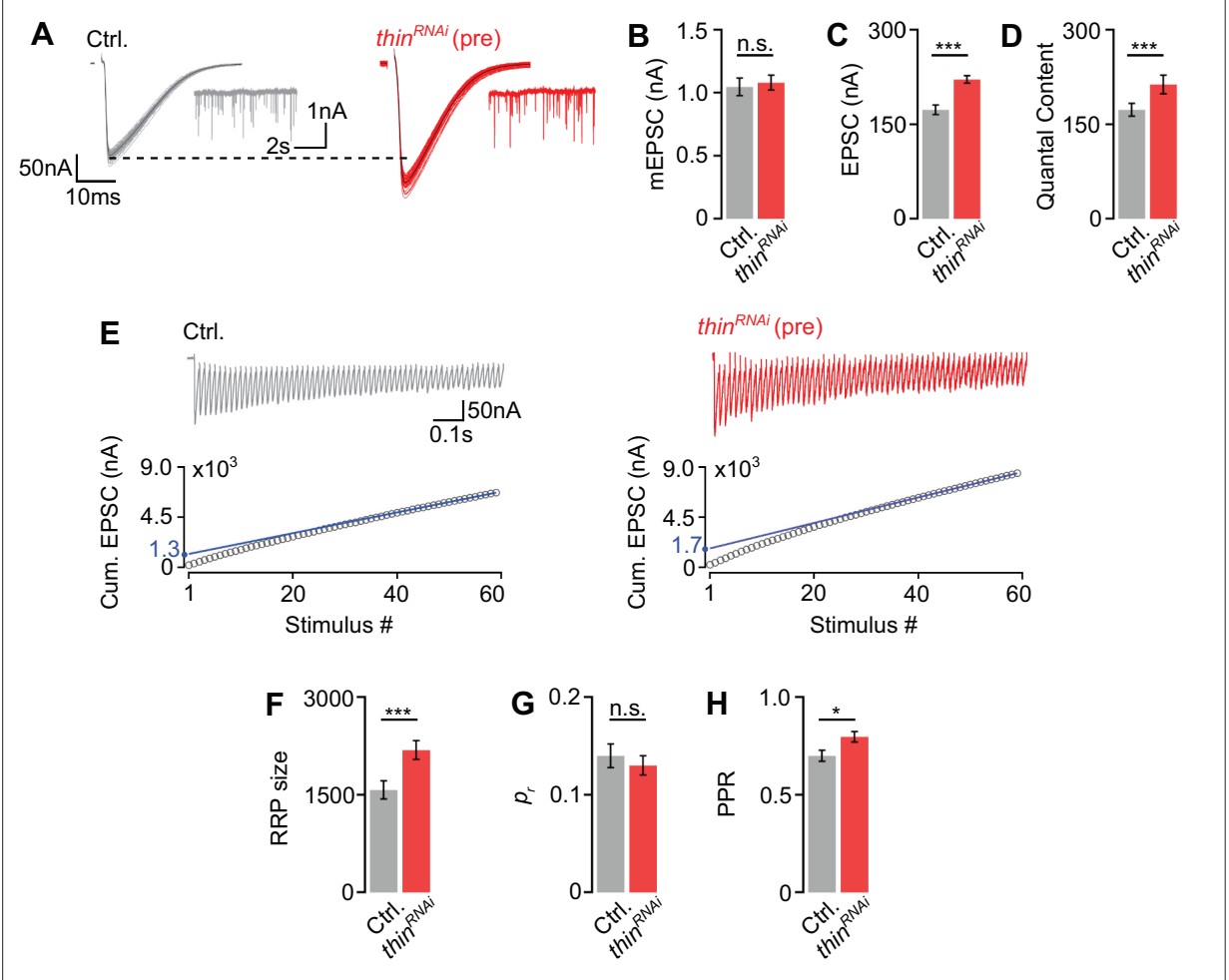

**Figure 4.** *Thin* negatively regulates release-ready vesicle number. (**A**) Representative excitatory postsynaptic currents (EPSCs) (individual sweeps and averages are shown in light colors and black, respectively), and mEPSCs (insets) of controls (*elav^c155^-Gal4>UAS-mCherry^RNAi^*, 'Ctrl.', gray) and presynaptic *thin^RNAi^* (*elav^c155^-Gal4>UAS-thin^RNAi^*, '*thin^RNAi^* (pre)', red). Mean mEPSC amplitudes (**B**), EPSC amplitudes (**C**), and quantal content (**D**) of the indicated genotypes. (**E**) Representative EPSC train (60 Hz, 60 stimuli, top) and cumulative EPSC amplitudes ('cum. EPSC', bottom) of control and presynaptic *thin^RNAi^*. The blue line is a line fit to the last 15 cum. EPSC amplitudes that was back-extrapolated to *t* = 0 (see Materials and methods). Mean readilyreleasable vesicle pool (RRP) size (cum. EPSC/mEPSC) (**F**), release probability ('*p_r_*', EPSC1/cum. EPSC) (**G**), and paired-pulse ratio ('PPR', EPSC2/EPSC1) (**H**) of the indicated genotypes. Note the increase in EPSC amplitude and RRP size in presynaptic *thin^RNAi^*. Mean ± standard error of the mean (SEM); Ctrl.: *n* = 16, *thin^RNAi^*: *n* = 17; *p < 0.05; ***p < 0.001; n.s.: not significant; Student's *t*-test.

The online version of this article includes the following source data and figure supplement(s) for figure 4:

**Source data 1.** Related to *Figure 4*.

**Figure supplement 1.** Presynaptic *thin^RNAi^* expression blocks presynaptic homeostatic plasticity (PHP) and induces a slight increase in AZ number.

**Figure supplement 1—source data 1.** Related to *Figure 4—figure supplement 1*.

genotypes (*Figure 2*, *Figure 3—figure supplement 1F–K*). Conversely, Brp intensity was unchanged after presynaptic *thin^RNAi^* expression (*elav^c155^-Gal4>UAS-thin^RNAi^*), while PHP was blocked and baseline synaptic transmission enhanced (*Figure 4*, *Figure 4—figure supplement 1*). Collectively, these data suggest that the morphological changes seen after *thin* perturbation are separable from synaptic physiology. Thus, although we cannot rule out that changes in NMJ morphology contribute to the PHP defect or the increase in release in *thin* mutants, we consider this possibility unlikely (see Discussion).

### *thin* negatively regulates release-ready vesicle number

Having established that *thin* is required for acute and sustained PHP expression, we next explored the role of *thin* in the regulation of neurotransmitter release under baseline conditions. *thin^ΔA^* mutants

display increased neurotransmitter release in the absence of PhTX, and this increase in release is rescued by presynaptic *thin* expression (*Figure 2*). We also noted a decrease in mEPSC amplitude in *thin*$^{\Delta A}$ mutants (*Figure 2B*), which may confound conclusions regarding presynaptic *thin* function. We therefore focused our further analyses on the effects of presynaptic *thin*$^{RNAi}$ expression.

First, we tested PHP after presynaptic *thin*$^{RNAi}$ expression (*elav*$^{c155}$-*Gal4>UAS-thin*$^{RNAi}$) and observed a complete PHP block (*Figure 4—figure supplement 1F–I*), providing independent evidence for a role of presynaptic *thin* in PHP. To elucidate the mechanisms through which *thin* negatively modulates release under baseline conditions, we probed the size of the readily releasable pool of synaptic vesicles (RRP) and neurotransmitter release probability ($p_r$) after presynaptic *thin* perturbation (*Figure 4*). Presynaptic *thin*$^{RNAi}$ expression (*elav*$^{c155}$-*Gal4>UAS-thin*$^{RNAi}$) significantly increased EPSC amplitudes and quantal content (*Figure 4A, C, D*), with no significant effects on mEPSC amplitudes compared to controls (*elav*$^{c155}$-*Gal4>UAS-mCherry*$^{RNAi}$; *Figure 4A, B*), suggesting that presynaptic *thin* represses release, consistent with the data obtained from *thin*$^{\Delta A}$ mutants (*Figure 2*). Note that the smaller mEPSC and EPSC amplitudes under baseline conditions after postsynaptic *thin* rescue (*Figure 2B, C*) compared to *thin*$^{RNAi}$ (*Figure 4B, C*) are most likely due to non-endogenous postsynaptic Thin levels caused by *thin* overexpression in the *thin*$^{\Delta A}$ mutant background. Next, we estimated RRP size using cumulative EPSC amplitude analysis during high-frequency stimulation (60 Hz; *Weyhersmüller et al., 2011*; *Müller et al., 2012*; *Figure 4E*). This analysis revealed a significantly larger RRP size upon presynaptic *thin*$^{RNAi}$ expression compared to controls (*Figure 4E, F*), implying that presynaptic *thin* negatively regulates RRP size. We then estimated $p_r$ based on the ratio between the first EPSC amplitude of the stimulus train and the cumulative EPSC amplitude, and observed no significant $p_r$ differences between presynaptic *thin*$^{RNAi}$ and controls (*Figure 4G*). We noted that the paired-pulse ratio between the second and first EPSC amplitude during 60 Hz stimulation was slightly increased after presynaptic *thin*$^{RNAi}$ expression compared to controls (*Figure 4H*), implying a slight decrease in $p_r$. These data suggest that the increase in release after presynaptic *thin*$^{RNAi}$ expression is unlikely caused by an increase in $p_r$, and that presynaptic *thin*$^{RNAi}$ expression may even slightly decrease $p_r$. Presynaptic *thin*$^{RNAi}$ expression also slightly increased Brp number (*Figure 4—figure supplement 1*), which may contribute to the increase in release after presynaptic *thin*$^{RNAi}$ expression (see Discussion). However, our analysis of *thin*$^{\Delta A}$ mutants implies that changes in NMJ size unlikely underlie the defects in synaptic physiology after presynaptic loss of *thin* (*Figures 2 and 3*). Together, we conclude that presynaptic *thin* opposes release by limiting the number of release-ready synaptic vesicles with largely unchanged $p_r$.

## Thin localizes in proximity to Dysbindin

TRIM32, Thin's predicted human ortholog (*Figure 1—figure supplement 2*), ubiquitinates Dysbindin and targets it for degradation in cultured human cells (*Locke et al., 2009*). *dysbindin*, in turn, is required for PHP at the *Drosophila* NMJ (*Dickman and Davis, 2009*), and genetic evidence suggests that the UPS controls Dysbindin under baseline conditions and during PHP at the *Drosophila* NMJ (*Wentzel et al., 2018*). We therefore explored the relationship between Thin and Dysbindin. First, we investigated the localization of Thin in relation to Dysbindin within synaptic boutons (*Figure 5*). Previous studies suggest very low endogenous Dysbindin levels that preclude direct immunohistochemical analysis at the *Drosophila* NMJ (*Dickman and Davis, 2009*; *Wentzel et al., 2018*). However, presynaptic expression of a fluorescently tagged *dysbindin* transgene revealed that Dysbindin localizes in close proximity to synaptic vesicle markers (*Dickman and Davis, 2009*; *Figure 5—figure supplement 1A–C*). The localization of fluorescently tagged Dysbindin likely overlaps with the one of endogenous Dysbindin, as its presynaptic expression rescues the PHP defect in *dysbindin* mutants (*Dickman and Davis, 2009*). Although we observed anti-Thin fluorescence in close proximity to Brp (*Figure 5—figure supplement 1D, E*), *thin* expression in *Drosophila* muscles makes it difficult to distinguish between presynaptic and postsynaptic Thin (*LaBeau-DiMenna et al., 2012*; *Figure 5—figure supplement 1D, E*). This prompted us to analyze the localization of fluorescently tagged Thin, which we expressed presynaptically (*elav*$^{c155}$-*Gal4>UAS-thin*$^{mCherry}$), in relation to Dysbindin. Presynaptic Thin$^{mCherry}$ partially overlapped with presynaptic fluorescently tagged Dysbindin (*elav*$^{c155}$-*Gal4>UAS-dysb*$^{venus}$) at confocal resolution (*Figure 5A and B*). The localization of fluorescently tagged Thin also likely overlaps with endogenous Thin, because presynaptic *thin* expression restores PHP and synaptic transmission in *thin* mutants (*Figure 2*). As indicated by the line profile across a bouton (*Figure 5B*), Dysbindin and Thin

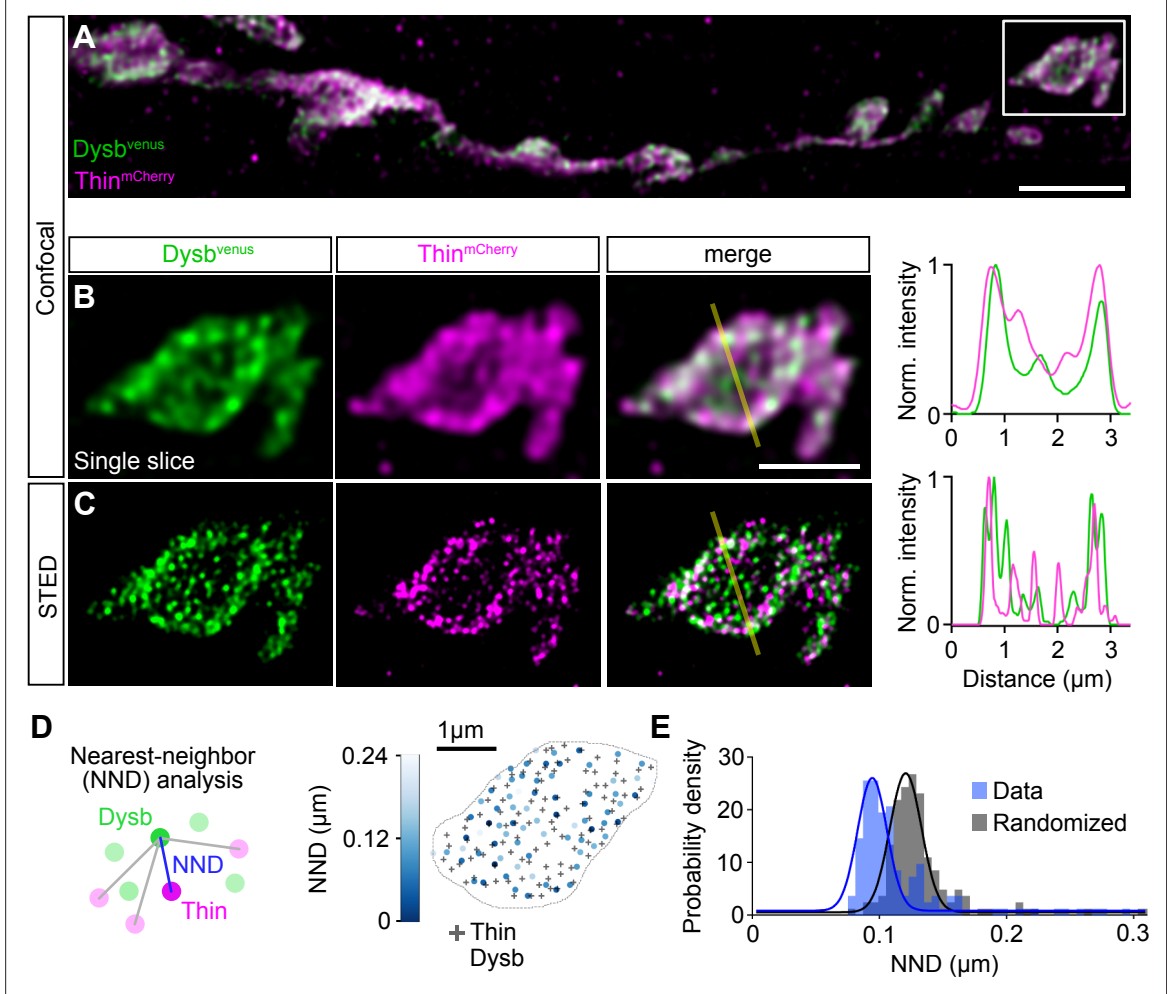

**Figure 5.** Thin localizes in close proximity to Dysbindin. (**A**) Confocal maximum intensity projection of a representative neuromuscular junction (NMJ) branch (muscle 6–7) after presynaptic coexpression (*elav^c155^-Gal4*) of venus-tagged Dysbindin (*UAS-venus-Dysbindin*, 'Dysb^venus^', green) and mCherry-tagged Thin (*UAS-mCherry-thin*, 'Thin^mCherry^', magenta) detected with anti-GFP and anti-DsRed, respectively. (**B**) Single plane of the synaptic bouton highlighted by the white square in (**A**) with corresponding line profile (right). The yellow line demarks the location of the line profile. (**C**) gSTED image of the synaptic bouton shown in (**B**) with corresponding line profile (right). Scale bar, A: 5 µm; B, C: 2 µm. Note the partial overlap between Thin^mCherry^ and Dysbindin^venus^ at confocal and STED resolution. (**D**) Left: Schematic of nearest-neighbor (NND) analysis between Thin^mCherry^ and Dysbindin^venus^ puncta at STED resolution. Right: Thin^mCherry^ puncta ('+', maximum locations, see Materials and methods) and the NNDs and locations of Dysbindin^venus^ puncta (color code denotes NND) of a representative bouton. (**E**) Histogram of mean Thin^mCherry^ − Dysbindin^venus^ NND per bouton of the recorded gSTED data (blue), or after randomized punctum distribution (gray, see Materials and methods). N = 10 NMJs, average n = 13 boutons per NMJ for data and simulations. Observed vs. randomized NNDs, p < 0.001; Student's *t*-test.

The online version of this article includes the following source data and figure supplement(s) for figure 5:

**Source data 1.** Related to *Figure 5*.

**Figure supplement 1.** Dysbindin and Synapsin distribute in the periphery of synaptic boutons, endogenous Thin localizes close to Brp, and presynaptic *dysbindin* overexpression does not affect neuromuscular junction (NMJ) morphology.

**Figure supplement 1—source data 1.** Related to *Figure 5—figure supplement 1F*.

**Figure supplement 2.** Thin localizes in close proximity to Dysbindin and Thin degrades Dysbindin in *Drosophila* S2 cells.

**Figure supplement 2—source data 1.** Related to *Figure 5—figure supplement 2*.

fluorescence intensity increased toward the bouton periphery (*Figure 5B*), similar to synaptic vesicle markers, such as synapsin (*Figure 5—figure supplement 1A, B*). With stimulated emission depletion microscopy with time-gated detection (gSTED), fluorescently tagged Thin and Dysbindin appeared as distinct spots (*Figure 5C*). To investigate the relationship between fluorescently tagged Thin and Dysbindin, we quantified the nearest-neighbor distance (NND) between Thin and Dysbindin spots

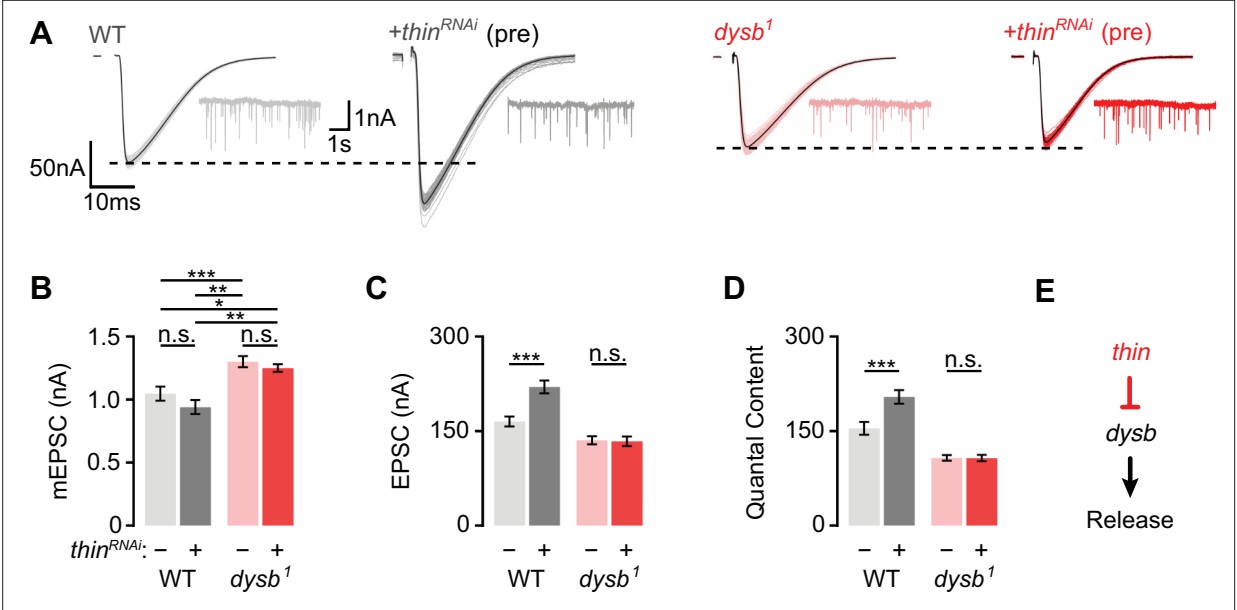

**Figure 6.** *Thin* represses release through *dysbindin*. (**A**) Representative excitatory postsynaptic currents (EPSCs) (individual sweeps and averages are shown in light colors and black, respectively), and mEPSCs (insets) of wild-type (WT) (gray) and presynaptic *thin^RNAi* (*elav^c155-Gal4>UAS-thin^RNAi*, '+*thin^RNAi* (pre)', dark gray), *dysb^1* mutants (light red), and presynaptic *thin^RNAi* in the *dysb^1* mutant background (*elav^c155-Gal4/Y*; *UAS-thin^RNAi/+*; *dysb^1*, '+*thin^RNAi* (pre)', dark red). Mean mEPSC amplitudes (**B**), EPSC amplitudes (**C**), and quantal content (**D**) of the indicated genotypes. Note that presynaptic *thin^RNAi* expression increases EPSC amplitude and quantal content in WT, but not in *dysb^1* mutants. Mean ± standard error of the mean (SEM); WT: $n = 17$, *elav^c155-Gal4>UAS-thin^RNAi*: $n = 17$, *dysb^1*: $n = 12$, *elav^c155-Gal4/Y*; *UAS-thin^RNAi/+*; *dysb^1*: $n = 12$; *p < 0.05; **p < 0.01; ***p < 0.001; n.s.: not significant; two-way analysis of variance (ANOVA) followed by Tukey's post hoc test. (**E**) Working model: Our genetic data support a model in which *thin* controls neurotransmitter release ('Release') through negative regulation of *dysbindin* ('*dysb*').

The online version of this article includes the following source data for figure 6:

**Source data 1.** Related to **Figure 6**.

---

(**Figure 5D**; see Materials and methods). This analysis revealed significantly smaller NNDs between Thin^mCherry and Dysb^venus spots than expected from random spot distributions (**Figure 5E**), implying a relationship between Thin and Dysbindin localization within synaptic boutons. Based on the proximity between Dysbindin and synaptic vesicle markers (**Dickman and Davis, 2009**; **Figure 5—figure supplement 1A–C**), these data indicate that a fraction of Thin localizes in the vicinity of Dysbindin and synaptic vesicles.

To provide independent evidence for a relationship between Thin and Dysbindin localization, and to explore if Thin acts as an E3 ubiquitin ligase for Dysbindin in *Drosophila*, we turned to cultured *Drosophila* Schneider 2 (S2) cells. Interestingly, while anti-Thin fluorescence was homogenously distributed within S2 cells under control conditions (**Figure 5—figure supplement 2A**), *dysbindin* (*dysb^venus*) overexpression led to a redistribution of anti-Thin fluorescence into clusters that localized in close proximity to anti-Dysbindin clusters (**Figure 5—figure supplement 2A**). Moreover, anti-Thin and anti-Dysbindin fluorescence intensities were highly correlated (**Figure 5—figure supplement 2B**), suggesting a possible interaction between Thin and Dysbindin in S2 cells, similar to the *Drosophila* NMJ (**Figure 5**). Next, we assessed whether Thin expression affects Dysbindin abundance in S2 cells by western blot analysis. We observed a decrease in Dysb^venus levels upon increasing Thin^HA expression levels (**Figure 5—figure supplement 2C, D**). Although we cannot exclude the possibility that Thin overexpression induced artificial Dysbindin ubiquitination by excess enzyme binding with low affinity, these data are consistent with the idea that Thin acts as an E3 ligase for Dysbindin in *Drosophila*, similar to TRIM32 in humans (**Locke et al., 2009**).

### *thin* represses release through *dysbindin*

We next explored a possible genetic interaction between *thin* and *dysbindin* in the context of synaptic physiology. As *thin* and *dysbindin* mutants alone disrupt PHP, the analysis of double mutants would

not be informative. We therefore investigated baseline synaptic transmission after presynaptic *thin*-*RNAi* expression in the *dysbindin* mutant background (*Figure 6*). Neither presynaptic *thin*RNAi expression (*elav*c155*-Gal4>UAS-thin*RNAi) in the WT background, nor in the *dysb*1 mutant background affected mEPSC amplitude (*Figure 6A, B*). While presynaptic *thin*RNAi expression enhanced EPSC amplitude and quantal content in the WT background (*Figure 6C, D*; see also *Figure 4*), presynaptic *thin*RNAi expression neither affected EPSC amplitude (*Figure 6C*) nor quantal content (*Figure 6D*) in the *dysb*1 mutant background. These data provide genetic evidence that *thin* negatively controls release through *dysbindin* (*Figure 6E*).

## Discussion

Employing an electrophysiology-based genetic screen targeting 157 E3 ligase-encoding genes at the *Drosophila* NMJ, we discovered that a mutation in the E3 ligase-encoding gene *thin* disrupts acute and sustained PHP. Presynaptic loss of *thin* led to increased release and RRP size, largely independent of gross synaptic morphological changes. Thin and Dysbindin localize in proximity within synaptic boutons, and biochemical evidence suggests that Thin degrades Dysbindin in vitro. Finally, presynaptic *thin* perturbation did not enhance release in the *dysbindin* mutant background, providing genetic evidence that *thin* represses release through *dysbindin*. As *thin* and *dysbindin* are required for PHP, these data are consistent with a model in which *thin* controls neurotransmitter release during PHP and under baseline conditions through *dysbindin*.

Our study represents the first systematic investigation of E3 ligase function in the context of synaptic transmission. A considerable fraction of the transgenic lines tested (11%) displayed a decrease in EPSC amplitude after PhTX treatment (*Figure 1C–E*). These E3 ligase-encoding genes may either be required for PHP and/or baseline synaptic transmission. Previous PHP screens in the same system identified PHP mutants with a hit rate of ~3% (*Dickman and Davis, 2009*; *Müller et al., 2011*). Thus, our data indicate that E3 ligase function plays a special role in PHP and/or baseline synaptic transmission. As more transgenic or mutant lines exhibited a decrease in synaptic transmission, we conclude that the net effect of E3 ligases is to promote synaptic transmission at the *Drosophila* NMJ. Given the evolutionary conservation of most E3 ligase-encoding genes tested in this study (*Figure 1*, *Supplementary file 1*), the results of our screen may allow predicting the roles of the tested E3 ligases in neurotransmitter release regulation in other systems.

Previous studies linked E3 ligases to synaptic development and synaptic function at the *Drosophila* NMJ (*Wan et al., 2000*; *van Roessel et al., 2004*). For instance, the E3 ligase *highwire* (*hiw*) restrains synaptic growth and promotes evoked synaptic transmission at the *Drosophila* NMJ (*Wan et al., 2000*). Similarly, the deubiquitinating protease *fat facets* represses synaptic growth and enhances synaptic transmission (*DiAntonio et al., 2001*). Although different molecular pathways have been implicated in *hiw*-dependent regulation of synaptic growth and function (*Russo et al., 2019*), it is generally difficult to disentangle effects on synaptic morphology from synaptic function. *thin* and its mammalian ortholog *TRIM32* are required for maintaining the cytoarchitecture of muscle cells (*Kudryashova et al., 2005*; *LaBeau-DiMenna et al., 2012*; *Cijsouw et al., 2018*). Hence, the changes in synaptic transmission described in the present study may be a secondary consequence of impaired muscle structure. However, presynaptic *thin* expression in the *thin* mutant background restored presynaptic function under baseline conditions and during homeostatic plasticity (*Figure 2*). Conversely, while postsynaptic *thin* expression largely rescued the defects in muscle morphology in *thin* mutants, the defects in synaptic function persisted. These genetic data suggest that the neurotransmitter release impairment under baseline condition and during PHP in *thin* mutants is unlikely caused by muscular dystrophy.

We also noted a slight increase in NMJ size and/or Brp number after postsynaptic rescue in the *thin* mutant background (*Figure 3*) or following presynaptic *thin*RNAi expression (*Figure 4—figure supplement 1*). Moreover, Brp intensity was decreased in *thin*ΔA mutants, after presynaptic rescue (*Figure 3*), or after postsynaptic *thin* overexpression in WT (*Figure 3—figure supplement 1*). The reasons for the increase in NMJ size or the decrease in Brp intensity after *thin* manipulations are unknown, but point at a potential dysregulation of *thin*-dependent pathways regulating NMJ size and Brp abundance. In principle, the changes in synaptic physiology observed in these genotypes may be a consequence of altered NMJ morphology. However, the observed changes in NMJ morphology were separable from changes in synaptic physiology (*Figures 2 and 3*,

*Figure 3—figure supplement 1*), implying that presynaptic *thin* regulates neurotransmitter release under baseline conditions and during homeostatic plasticity largely independent of changes in synaptic morphology.

We revealed that presynaptic *thin* perturbation results in enhanced neurotransmitter release (*Figures 2, 4 and 6*), indicating that the E3 ligase Thin represses neurotransmitter release under baseline conditions. Notably, there are just a few molecules that have been implicated in repressing neurotransmitter release, such as the SNARE-interacting protein tomosyn (*Hatsuzawa et al., 2003*; *Chen et al., 2011*), or the RhoGAP crossveinless-c (*Pilgram et al., 2011*). How could the E3 ligase Thin oppose neurotransmitter release? We discovered that *dysbindin* is required for the increase in release induced by presynaptic *thin* perturbation (*Figure 6*). Moreover, Thin localizes in close proximity to Dysbindin in synaptic boutons (*Figure 5*), and we provide evidence that Thin likely degrades Dysbindin in vitro (*Figure 5—figure supplement 2*), similar to its mammalian ortholog TRIM32 (*Locke et al., 2009*). At the *Drosophila* NMJ, 26S-proteasomes are transported to presynaptic boutons (*Kreko-Pierce and Eaton, 2017*), where they degrade proteins on the minute time scale (*Speese et al., 2003*; *Wentzel et al., 2018*). Previous genetic data suggest a positive correlation between Dysbindin levels and neurotransmitter release (*Dickman and Davis, 2009*; *Wentzel et al., 2018*), and there is genetic evidence for rapid, UPS-dependent degradation of Dysbindin at the *Drosophila* NMJ (*Wentzel et al., 2018*). In combination with these previous observations, our data are consistent with the idea that Thin opposes release by acting on Dysbindin. Although the low abundance of endogenous Dysbindin at the *Drosophila* NMJ precludes direct analysis of Dysbindin levels (*Dickman and Davis, 2009*), we speculate that Thin decreases Dysbindin abundance by targeting it for degradation. Alternatively, Thin may modulate Dysbindin function through mono-ubiquitination. Genetic data suggest that Dysbindin interacts with the SNARE protein SNAP25 through Snapin (*Dickman et al., 2012*). Hence, Thin-dependent regulation of Dysbindin may modulate release via Dysbindin's interaction with the SNARE complex.

Our study identified a crucial role for *thin* in PHP. How does the increase in neurotransmitter release in *thin* mutants under baseline conditions relate to the PHP defect? mEPSC amplitudes were decreased in *thin* mutants, after presynaptic and postsynaptic rescue (*Figure 2*), and largely unchanged after presynaptic *thin*^*RNAi* expression (*Figure 4*). The decrease in mEPSC amplitude implies a postsynaptic role of *thin* in regulating quantal size, possibly by regulating GluR levels (*Figure 2—figure supplement 1*). Quantal content was increased in *thin* mutants, after postsynaptic rescue (*Figure 2*), and after presynaptic *thin*^*RNAi* expression (*Figure 4*). Together, these data imply that the increase in quantal content under baseline conditions induced by presynaptic *thin* manipulations is separable from a decrease in miniature amplitude. By definition, PHP is induced by a relative decrease in miniature amplitude. Given that quantal content increased after presynaptic *thin* manipulations independent of changes in miniature amplitude, we consider it unlikely that the increased quantal content under baseline conditions represents a homeostatic response. Could the increase in release after presynaptic *thin* perturbation simply occlude PHP? The relative increase in release during PHP of WT synapses exceeds the increase in release under baseline conditions in *thin* mutants. Thus, although we cannot exclude that PHP is solely occluded by enhanced baseline release in *thin* mutants, we consider this scenario unlikely.

PHP is blocked by acute pharmacological, or prolonged genetic proteasome perturbation at the *Drosophila* NMJ (*Wentzel et al., 2018*). Moreover, PHP at this synapse requires *dysbindin* (*Dickman and Davis, 2009*), and genetic data suggest UPS-dependent control of a Dysbindin-sensitive vesicle pool during PHP (*Wentzel et al., 2018*). Based on our finding that *thin* is required for acute and sustained PHP expression (*Figure 2*, *Figure 2—figure supplement 1*), and the links between *thin* und *dysbindin* in the context of release modulation outlined above, we propose a model in which Thin-dependent ubiquitination of Dysbindin is decreased during PHP. Given the positive correlation between Dysbindin levels and release (*Dickman et al., 2012*; *Wentzel et al., 2018*), the resulting increase in Dysbindin abundance would potentiate release. Further work is needed to test how Thin is regulated during PHP. Thin is the first E3 ubiquitin ligase linked to homeostatic regulation of neurotransmitter release. Interestingly, a recent study revealed a postsynaptic role for Insomniac, a putative adaptor of the Cullin-3 ubiquitin ligase complex, in PHP at the *Drosophila* NMJ (*Kikuma et al., 2019*), suggesting a key function of the UPS in both synaptic compartments during PHP at this synapse.

TRIM32, the human ortholog of *thin*, is required for synaptic down-scaling in cultured hippocampal rat neurons (*Srinivasan et al., 2020*), as well as long-term potentiation in hippocampal mouse brain slices (*Ntim et al., 2020*), implying a broader role of this E3 ubiquitin ligase in synaptic plasticity. TRIM32 has been implicated in various neurological disorders, such as depression (*Ruan et al., 2014*), Alzheimer's disease (*Yokota et al., 2006*), autism spectrum disorder (*Lionel et al., 2014*; *Ruan et al., 2014*), or attention deficit hyperactivity disorder (*Lionel et al., 2011*). It will be exciting to explore potential links between TRIM32-dependent control of synaptic homeostasis and these disorders in the future.

# Materials and methods

## Key resources table

| Reagent type (species) or resource | Designation | Source or reference | Identifiers | Additional information |
|---|---|---|---|---|
| Genetic reagent (*Drosophila melanogaster*) | *thin*$^{\Delta A}$ | *LaBeau-DiMenna et al., 2012* | | |
| Genetic reagent (*Drosophila melanogaster*) | *UAS-abba* | *LaBeau-DiMenna et al., 2012* | | |
| Genetic reagent (*Drosophila melanogaster*) | *UAS-mCherry-thin* | This study | | Stock is available upon request |
| Genetic reagent (*Drosophila melanogaster*) | *GluRIIA*$^{SP16}$ | *Petersen et al., 1997* | | |
| Genetic reagent (*Drosophila melanogaster*) | *dysbindin*$^{1}$ | *Dickman and Davis, 2009* | | |
| Genetic reagent (*Drosophila melanogaster*) | *UAS-thin*$^{RNAi}$ | *Perkins et al., 2015* | RRID:BDSC_42826 | |
| Genetic reagent (*Drosophila melanogaster*) | *UAS-mCherry*$^{RNAi}$ (P{VALIUM20-mCherry}attP2) | Bloomington *Drosophila* Stock Center | RRID:BDSC_35785 | |
| Genetic reagent (*Drosophila melanogaster*) | *UAS-venus-dysbindin* | *Dickman and Davis, 2009* | | |
| Genetic reagent (*Drosophila melanogaster*) | *elav*$^{c155}$-*Gal4* | Bloomington *Drosophila* Stock Center | RRID:BDSC_458 | |
| Genetic reagent (*Drosophila melanogaster*) | *24B-Gal4* | Bloomington *Drosophila* Stock Center | RRID:BDSC_1767 | |
| Antibody | anti-Bruchpilot (nc82) (mouse monoclonal) | DSHB, University of Iowa, USA | RRID:AB_2314866 | (1:100) |
| Antibody | anti-GFP (rabbit polyclonal) | Thermo Fisher Scientific | Thermo Fisher Scientific Cat# G10362, RRID:AB_2536526 | IF: (1:500) WB: (1:1000) |
| Antibody | anti-GFP (mouse mono clonal) | Thermo Fisher Scientific | Thermo Fisher Scientific Cat# A-11120, RRID:AB_221568 | (1:500) |
| Antibody | anti-DsRed (mouse monoclonal) | Santa Cruz Biotechnology | Santa Cruz Biotechnology Cat# sc-390909, RRID:AB_2801575 | (1:500) |
| Antibody | anti-SYNORF1 (Synapsin, 3C11) (mouse monoclonal) | DSHB, University of Iowa, USA | RRID:AB_528479 | (1:250) |
| Antibody | anti-Thin (guinea pig polyclonal) | *LaBeau-DiMenna et al., 2012* | | Larva: (1:200) S2: (1:400) |

*Continued on next page*

*Continued*

| Reagent type (species) or resource | Designation | Source or reference | Identifiers | Additional information |
|---|---|---|---|---|
| Antibody | anti-HRP Alexa-Fluor 647 (goat polyclonal) | Jackson ImmunoResearch Labs | Jackson ImmunoResearch Labs Cat# 123-605-021, RRID:AB_2338967 | (1:200) |
| Antibody | Anti-HA (mouse monoclonal) | Biolegend | BioLegend Cat# 901533, RRID:AB_2801249 | (1:1000) |
| Antibody | Anti-BetaTubulin (mouse monoclonal) | DSHB, University of Iowa, USA | DSHB Cat# E7, RRID:AB_528499 | (1:1000) |
| Antibody | Goat anti-Mouse IgG (H+L) Secondary Antibody, HRP (goat polyclonal) | Thermo Fisher Scientific | Thermo Fisher Scientific Cat# 31430, RRID:AB_228307 | (1:2000) |
| Antibody | Goat anti-Rabbit IgG (H+L) Secondary Antibody, HRP (goat polyclonal) | Thermo Fisher Scientific | Thermo Fisher Scientific Cat# 32460, RRID:AB_1185567 | (1:2000) |
| Antibody | Alexa-Fluor anti-mouse 488 (goat polyclonal) | Thermo Fisher Scientific | Thermo Fisher Scientific Cat# A32723, RRID:AB_2633275 | (1:500) |
| Antibody | Alexa Fluor anti-guinea pig 555 (goat polyclonal) | Thermo Fisher Scientific | Thermo Fisher Scientific Cat# A-21435 RRID:AB_2535856 | (1:400) |
| Antibody | Atto 594 conjugated anti-mouse (goat polyclonal) | Sigma-Aldrich | Sigma-Aldrich Cat# 76,085 | (1:100) |
| Antibody | Abberior STAR 635P (goat polyclonal) | Abberior | Abberior Cat# ST635P-1002-500 UG, RRID:AB_2893229 | (1:100) |
| Chemical compound, drug | Bouin's fixative | Sigma-Aldrich | HT-10132 | |
| Chemical compound, drug | Ethanol | Merck | CAS# 64-17-5 | |
| Chemical compound, drug | ProLong Gold Antifade | Thermo Fisher Scientific | P36930 | |
| Chemical compound, drug | Philanthotoxin-433 | Santa Cruz Biotechnology | Cat# sc-255421 | |
| Chemical compound, drug | Schneider's *Drosophila* medium | Gibco | Cat# 21720024 | |
| Chemical compound, drug | FuGENE HD | Promega | Cat# E2311 | |
| Chemical compound, drug | Paraformaldehyde | Merck | HT501128 | |
| Chemical compound, drug | NP-40 | Thermo Fisher Scientific | 85,125 | |
| Chemical compound, drug | Deoxycholate | Sigma-Aldrich | D6750 | |
| Chemical compound, drug | cOmplete | Sigma-Aldrich | 11697498001 | |
| Chemical compound, drug | ECL Prime Western Blotting Detection Reagent | GE Healthcare | Cat# 28980926 | |
| Cell line (*D. melanogaster*) | *Drosophila* Schneider 2 (S2) Cells | Thermo Fisher Scientific | Cat# R69007 | |
| Commercial assay, kit | Nitrocellulose membrane | Amersham Hibond GE Healthcare | Cat# 88,018 | |

*Continued on next page*

*Continued*

| Reagent type (species) or resource | Designation | Source or reference | Identifiers | Additional information |
|---|---|---|---|---|
| Recombinant DNA reagent | pMT-Gal4 | Addgene | RRID:Addgene_53366 | |
| Software, algorithm | Fiji / ImageJ | https://fiji.sc | RRID:SCR_002285 | Version 1.51n |
| Software, algorithm | Clampex | Axon CNS, Molecular Devices | RRID:SCR_011323 | |
| Software, algorithm | Leica Application Suite X | Leica Microsystems | RRID:SCR_013673 | |
| Software, algorithm | Huygens Software | https://svi.nl/HuygensSoftware | RRID:SCR_014237 | |
| Software, algorithm | Igor Pro | WaveMetrics | RRID:SCR_000325 | Version 6.37 |
| Software, algorithm | NeuroMatic | *Rothman and Silver, 2018* | RRID:SCR_004186 | Version 3.0c |
| Software, algorithm | NumPy | https://www.numpy.org | RRID:SCR_008633 | |
| Software, algorithm | SciPy | https://www.scipy.org | RRID:SCR_008058 | |
| Software, algorithm | IPython | http://ipython.org | RRID:SCR_001658 | |
| Software, algorithm | Neo | http://neuralensemble.org/neo | RRID:SCR_000634 | |
| Software, algorithm | Shapely | (*Gillies, 2007*) https://github.com/shapely/shapely | | |
| Software, algorithm | RStudio | (*R Studio Team, 2020*) http://www.rstudio.com/ | RRID:SCR_000432 | Version 2021.09.0 |
| Software, algorithm | pwr-package | (*Champely, 2020*) https://github.com/heliosdrm/pwr | | |
| Software, algorithm | GNU Image Manipulation Program | https://www.gimp.org/ | RRID:SCR_003182 | Version 2.8.10 |
| Software, algorithm | Inkscape | http://www.inkscape.org | RRID:SCR_014479 | Version 0.92.2 |
| Software, algorithm | Affinity Designer | https://affinity.serif.com/en-us/designer/ | RRID:SCR_016952 | Version 1.10.4 |

## Fly stocks and genetics

*Drosophila* stocks were maintained at 21–25°C on normal food. The $w^{1118}$ strain was used as the WT control. *GluRIIA$^{SP16}$* mutants (*Petersen et al., 1997*) and *dysbindin$^1$* mutants (*Dickman and Davis, 2009*) were a kind gift from Graeme Davis' lab. *thin$^{ΔA}$* mutants and *UAS-abba* transgenic flies, now referred to as *UAS-thin* (*LaBeau-DiMenna et al., 2012*), were a generous gift from Erika Geisbrecht. The *UAS-thin$^{RNAi}$* stock (BDSC stock 42826, *Perkins et al., 2015*) and the *UAS-mCherry$^{RNAi}$* stock (BDSC stock 35785) were obtained from the Bloomington *Drosophila* Stock Center (BDSC, Bloomington, IN, USA), and the *UAS-venus-dysbindin* line was provided by Dion Dickman's lab. For pan-neuronal expression, the *elav$^{c155}$-Gal4* (on the X chromosome) driver line was used and analysis was restricted to male larvae. For expression in muscle cells, we used the *24B-Gal4* driver line. Both driver lines were obtained from the BDSC. Standard second and third chromosome balancer lines (BDSC) and genetic strategies were used for all crosses and for maintaining the mutant lines. For the generation of transgenic flies carrying *UAS-mCherry-thin*, constructs based on the pUAST-attB vector backbone were injected into the ZP-attP-86Fb fly line harboring a landing site on the third chromosome according to standard procedures (*Bischof et al., 2014*).

## Cell culture and transfection

Schneider S2 cells were obtained from Thermo Fisher Scientific ('Gibco *Drosophila* S2 cells'; Cat# R69007). The supplier's Master Seed Bank was characterized by isozyme and karyotype analysis, and was tested for contamination of bacteria, yeast, mycoplasma, and virus. We have not independently verified cell line identity or tested for mycoplasma contamination. However, contamination with other cell lines is unlikely, because the used cell line is (1) the only cell line used in the lab, (2) the only *Drosophila* cell line present at the institute, and (3) cells grow at 25° and in a different medium compared to human cell lines. Cells were used within 10 months after purchase. Schneider S2 cells

were cultivated in standard Schneider's *Drosophila* medium (Gibco) containing 10% fetal calf serum and 5% penicilin/streptomycin at 25°C. For immunohistochemistry and microscopy, cells were plated on cover slips in 12-well plates with 80% density and transfected with 1.5 μg (total) vector DNA using FuGENE HD Transfection Reagent according to the standard protocol. The following vectors were used: pMT-Gal4 (Addgene), pUAS-mCherry-thin, pUAS-HA-thin, pUAS-venus-dysbindin (Dion Dickman), and empty pUAS to adjust to equal DNA levels. Twenty-four hours after plating, $CuSO_4$ (0.5 mM) was added to the culture for 24 hr to induce the expression of the pMT vector driving Gal4, which in turn drives transcription of UAS constructs.

## Plasmid construction

For the pUAS-mCherry-thin vector, mCherry was cloned into pUAS-attB (Addgene) via EcoRI/NotI using the following primers:

(fw: 5'-GCGAATTCATGGTGAGCAAGGGCGAGGAG-3', rev: 5'- GCGCGGCCGCCCTTGTACAGCTCGTCCATGCCG-3').

Thin isoform A (NM_137546.3) was amplified from *Drosophila* cDNA by PCR using the following primers:

(fw: 5'-CGGCGGCCGCATGGAGCAATTCGAGCAGCTGT-3', rev: 5'-CGTCTAGAATGAAGACTTGGACGCGGTGATTCTCTCG-3') and then cloned into the pUAS-mCherry vector via NotI/XbaI.

pUAS-HA-thin was generated by In-Fusion mutagenesis (TaKaRa) from the pUAS-mCherry-thin plasmid with the following primers:

(fw: 5'-AGATTACGCTTATCCATATGATGTTCCAGATTATGCTGGCCGCATGGAGCAATTC-3' and rev: 5'-GGATAAGCGTAATCTGGAACATCGTATGGGTACATAATTCCCAATTCCCTATTCAGAGT-3').

Correct cloning was confirmed by sequencing on all final vectors.

## Electrophysiology

Electrophysiological recordings were made from third-instar larvae at the wandering stage. Larvae were dissected and sharp-electrode recordings were made from muscle 6 in abdominal segments 3 and 4 using an Axoclamp 900 A amplifier (Molecular Devices). The extracellular HL3 saline contained (in mM): 70 NaCl, 5 KCl, 10 $MgCl_2$, 10 Na-HEPES (N-2-hydroxyethylpiperazine-N'-2-ethanesulfonic acid), 115 sucrose, 5 trehalose, 5 HEPES, 1.5 $CaCl_2$. To induce PHP, larvae were incubated with 20 μM PhTX-433 (Santa Cruz Biotechnology) for 10 min at room temperature after partial dissection (see *Frank et al., 2006*). AP-evoked EPSCs were induced by stimulating hemi-segmental nerves with single APs (0.3-ms stimulus duration, 0.3 Hz), and recorded with a combination of a HS-9A × 10 and a HS-9A × 0.1 headstage (Molecular Devices) in two-electrode voltage clamp mode. mEPSPs and mEPSCs were recorded with one or two HS-9A × 0.1 headstage(s) (Molecular Devices), respectively. Muscle cells were clamped to a membrane potential of −65 mV for EPSC and −100 mV for mEPSC recordings to increase the signal-to-noise ratio.

A total of 50 EPSCs were averaged to obtain the mean EPSC amplitude for each NMJ. mEPSCs and EPSCs were recorded in different NMJs because different headstage combinations were used to improve the signal-to-noise-ratio for mEPSC recordings. Hence, quantal content was calculated by dividing the mean EPSC amplitude of each NMJ by the average of the average mEPSC amplitude of all NMJs of a given experimental group. RRP size was estimated by the method of cumulative EPSC amplitudes (*Schneggenburger et al., 1999*). NMJs were stimulated with 60 Hz trains (60 stimuli, 5 trains per cell), and the cumulative EPSC amplitude was obtained by back-extrapolating a linear fit to the last 15 cumulative EPSC amplitude values of the 60 Hz train to time zero. The cumulative EPSC amplitude of each NMJ was then divided by the average mEPSC amplitude of all NMJs of a given experimental group to obtain the RRP estimate.

## Immunohistochemistry and microscopy
### *Drosophila* NMJ

Third-instar larval preparations were fixed for 3 min with Bouin's fixative (100%, Sigma-Aldrich) for confocal microscopy, or ice-cold ethanol (100%, Merck) for 10 min for confocal/STED microscopy. Preparations were washed thoroughly with phosphate-buffered saline (PBS) containing 0.1% Triton X-100 (PBST). After washing, preparations were blocked with 3% normal goat serum in PBST. Incubation with the primary antibody was done at 4°C on a rotating platform overnight. The following

antibodies and dilutions were used for NMJ stainings: (*Primary*) anti-Bruchpilot (nc82, mouse, DSHB, 1:100), anti-GFP (rabbit, Thermo Fisher Scientific, 1:500), anti-GFP (mouse, Thermo Fisher Scientific, 1:500), anti-DsRed (mouse, Santa Cruz Biotechnology, 1:500), anti-SYNORF1 (Synapsin, 3C11, mouse, DSHB, 1:250), anti-Thin (guinea pig, gift from Erika R. Geisbrecht, 1:200), and anti-HRP Alexa-Fluor 647 (goat, Jackson ImmunoResearch, 1:200). For confocal microscopy, Alexa-Fluor anti-mouse 488 (Thermo Fisher Scientific; 1:500) and Alexa Fluor anti-guinea pig 555 (Thermo Fisher Scientific; 1:400) were applied overnight at 4°C on a rotating platform. For gSTED microscopy, the following secondary antibodies (1:100) were applied for 2 hr at room temperature (RT) on a rotating platform: Atto 594 (anti-mouse, Sigma-Aldrich) and Abberior STAR 635P (anti-rabbit, Abberior). Experimental groups of a given experiment were processed in parallel in the same tube. Preparations were mounted onto slides with ProLong Gold (Thermo Fisher Scientific).

## S2 cell culture

S2 cells grown on coverslips were washed with PBST and fixed with 10% PFA (paraformaldehyde) for 10 min. After washing three times with PBST, preparations were blocked with 5% normal goat serum in PBST for 30 min. Incubation with primary antibodies was done at RT on a rotating platform for 2 hr. The following antibodies were used for S2 cell stainings: anti-thin (guinea pig, gift from Erika R. Geisbrecht, 1:400), anti-Dysbindin (mouse, gift from Dion Dickman, 1:400). After washing three times with PBST, cells were incubated with the secondary antibodies Alexa Fluor anti-guinea pig 555 and Alexa Fluor anti-mouse 488 (Thermo Fisher Scientific; both 1:400) at RT on a rotating platform for 2 hr. Cover slips were mounted onto slides with ProLong Gold (Thermo Fisher Scientific) after three PBST washes.

## Confocal and gSTED microscopy

Images were acquired with an inverse Leica TCS SP8 STED 3× microscope (Leica Microsystems, Germany) of the University of Zurich Center for Microscopy and Image Analysis. Excitation light (580 or 640 nm) of a flexible white light laser was focused onto the specimen using a 100× objective (HC PL APO 1.40 NA Oil STED WHITE; Leica Microsystems, Germany) with immersion oil conforming to ISO 8036 with a diffraction index of $n$ = 1.5180 (Leica Microsystems, Germany). For gSTED imaging, the flexible white light laser was combined with a 775 nm STED depletion laser. Emitted light was detected with two HyD detectors in photon counting mode (Leica Microsystems, Germany). Pixel size was 20 × 20 nm and z-stacks were acquired with a step size of 120 nm. For STED imaging, we used time-gated single photon detection (empirical adjustment within a fluorescence lifetime interval from 0.7 to 6.0 ns). Line accumulation was set to 1 and 6 for confocal and STED imaging, respectively. Images were acquired with Leica Application Suite X software (Leica Application Suite X, version 2.0; Leica Microsystems, Germany). Experimental groups were imaged side-by-side with identical settings.

Images were processed and deconvolved with Huygens Professional (Huygens compute engine 17.04, Scientific Volume Imaging B.V., Netherlands). In brief, the 'automatic background detection' tool (radius = 0.7 μm), and the 'auto stabilize' feature were used to correct for background and lateral drift. Images were deconvolved using the Good's roughness Maximum Likelihood algorithm with default parameter settings (maximum iterations: 10; signal-to-noise ratio: 7 for STED and 15 for confocal; quality threshold: 0.003).

## Western blot

Transfected cells in 12-well plates were harvested after 72 hr, washed with PBS and lysed by adding 50 μl of RIPA buffer (50 mM Tris, pH 8.0, 150 mM NaCl, 1% Nonidet P-40, 0.5% deoxycholate, 0.1% sodium dodecyl sulfate (SDS), 0.4 mM EDTA (ethylenediaminetetraacetic acid), 10% glycerol) containing protease inhibitors (cOmplete, Mini, EDTA-free Protease Inhibitor Cocktail, Sigma-Aldrich) for 30 min on ice. The lysates were sonified three times for 1 min and boiled for 5 min in SDS-sample buffer containing 5% β-mercaptoethanol. Samples were separated on acrylamide gels using SDS–polyacrylamide gel electrophoresis (PAGE), then transferred to nitrocellulose membranes (Amersham Hibond GE Healthcare). After blocking in 5% milk in PBST for 1 hr, membranes were incubated in the following primary antibodies: anti-GFP (rabbit, Thermo Fisher Scientific, 1:1000), anti-HA (mouse, Biolegend, 1:1000), and anti-Tubulin (E7, mouse, DSHB, 1:1000) in blocking solution overnight. Horseradish peroxidase-conjugated secondary antibodies (anti-mouse-HRP and anti-rabbit-HRP, 1:2000 in blocking solution) were applied to membranes for 2 hr. Detection was performed using ECL Reagent

(GE Healthcare, Chicago, IL, USA). Western blots were revealed using enhanced chemiluminescence and imaged using a Fusion FX7 system (Vilber Lourmat). Densitometric analyses (mean pixel intensity of a ROI containing a band of interest) were done in Fiji/ImageJ.

## Data analysis

### Electrophysiology

Electrophysiology data were acquired with Clampex (Axon CNS, Molecular Devices) and analyzed with custom-written routines in Igor Pro (Wavemetrics). For the genetic screen data, mEPSPs were detected with a template matching algorithm implemented in Neuromatic (Rothman & Silver, 2018) running in Igor Pro (Wavemetrics). The average mEPSP amplitude was calculated from all detected events in a recording after visual inspection for false positives. For the remaining data, mEPSC data were analyzed using routines written with scientific python libraries, including numpy, scipy, IPython and neo (*Garcia et al., 2014*), and mEPSCs were detected using an implementation of a template-matching algorithm (*Clements and Bekkers, 1997*).

### NMJ morphology

Microscopy images were analyzed using custom-written routines in ImageJ/Fiji (version 1.51n, National Institutes of Health, USA). Brp quantification was performed as follows: First, individual Brp puncta were isolated by segmenting binary fluorescence intensity threshold masks (15% or 35% of the maximum intensity value) of background corrected (rolling ball, radius = 1 μm) and filtered (3 × 3 median) maximum intensity projection images. The number of Brp objects in the mask served as a proxy for AZ number and was normalized to the area of the HRP mask (binary mask, 15% or 35% of the maximum intensity value). Average Brp-intensity values were calculated for each Brp punctum from background-corrected, unfiltered maximum intensity projection images.

### NND analysis

For the NND analysis (*Figure 5*), individual synaptic boutons were segmented manually within a deconvolved gSTED stack and a single plane in the middle of the bouton was extracted for further analysis. Next, Fiji's 'Find Maxima' algorithm was used to obtain the $x,y$ coordinate of the brightest pixel within each Dysbindin$^{venus}$ and Thin$^{mCherry}$ punctum. For the maximum of each Dysbindin punctum, the distances to the maxima of all Thin puncta within the same bouton were measured and the NNDs were calculated. For each bouton, the analysis was repeated after assigning random $x,y$ coordinates to each Dysbindin and Thin punctum within the bouton boundaries using the Python package Shapely (Gillies and others, 2007; https://github.com/shapely/shapely). NND values were averaged for each bouton.

### Correlation analysis S2 cells

Pearson's correlation coefficients ($r$) were calculated for each pixel in single confocal planes of *Drosophila* S2 cells coexpressing Dysbindin$^{venus}$ and Thin$^{mCherry}$ using Costes' approach (*Costes et al., 2004*) implemented in the JACoP toolbox of ImageJ/Fiji (*Bolte and Cordelières, 2006*; *Figure 5— figure supplement 2B*). The algorithm also creates simulated images by randomly sampling point spread function-sized chunks of the original image, and calculating $r$ for each pixel of the simulated data.

### Statistics

Statistical analyses were done using RStudio Team (2021). RStudio: Integrated Development Environment for R. RStudio, PBC, Boston, MA, http://www.rstudio.com/. For more than two factors, we used two-way analysis of variance (ANOVA) followed by Tukey's post hoc test to correct for multiple comparisons between genotypes and conditions. For one factor with more than two groups, one-way ANOVA with Tukey's multiple comparisons was performed. Two-sided Student's $t$-tests or nonparametric Mann–Whitney $U$-tests were used for comparison between two groups after a Shapiro–Wilk test and a Levene's test. Statistical significance (p) was set to 0.05 (*), 0.01 (**), and 0.001 (***). Power analysis was performed using the pwr-package of Rstudio. Minimum desired effect size based on Cohen's d value was used to estimate the minimum sample size for a power ≥0.8 and a significance

level of 0.05 for two-sided Student's *t*-tests or Mann–Whitney *U*-tests. Data are given as mean ± standard error of the mean (SEM).

Figures were assembled using GIMP (The GIMP team, 2.8.10, https://www.gimp.org/), Inkscape (Inkscape project, 0.92.2. http://www.inkscape.org), and Affinity Designer (1.10.4, Serif (Europe) Ltd, West Bridgford, Nottinghamshire, United Kingdom).

## Acknowledgements

We are grateful to the members of the Müller lab for helpful discussions and critical comments on the manuscript. We thank Dr. Damian Szklarczyk for help with STRING-based protein–protein interaction analysis used for prioritization of E3 ligases, and Dr. Marian Hruska-Plochan for help with the western blot analysis. Stocks obtained from the Bloomington *Drosophila* Stock Center (NIH P40OD018537) were used in this study.

## Additional information

### Funding

| Funder | Grant reference number | Author |
| --- | --- | --- |
| Schweizerischer Nationalfonds zur Förderung der Wissenschaftlichen Forschung | PP00P3-15 | Martin Müller |
| European Research Council | SynDegrade-679881 | Martin Müller |

The funders had no role in study design, data collection, and interpretation, or the decision to submit the work for publication.

### Author contributions

Martin Baccino-Calace, Conceptualization, Software, Formal analysis, Investigation, Visualization, Writing – original draft; Katharina Schmidt, Conceptualization, Formal analysis, Investigation, Visualization, Methodology; Martin Müller, Conceptualization, Resources, Data curation, Software, Formal analysis, Supervision, Funding acquisition, Validation, Investigation, Visualization, Methodology, Writing – original draft, Project administration, Writing – review and editing

### Author ORCIDs

Martin Baccino-Calace http://orcid.org/0000-0003-4143-4166
Katharina Schmidt http://orcid.org/0000-0003-2797-9952
Martin Müller http://orcid.org/0000-0003-1624-6761

### Decision letter and Author response

Decision letter https://doi.org/10.7554/eLife.71437.sa1
Author response https://doi.org/10.7554/eLife.71437.sa2

## Additional files

### Supplementary files

• Transparent reporting form

• Supplementary file 1. Summary table of electrophysiology data for the genetic screen. Data are mean ± SEM. UAS-RNAis were driven in neurons by elavc155-Gal4, and elavc155-Gal4/Y served as the control. We tested 157 putative E3 ligase-encoding genes and 11 E3-associated genes, using 180 lines (UAS-RNAi or mutants; some genes were targeted with multiple lines; mean *n* = 4 NMJs per line, range 3–12 per line). Control data were continuously collected throughout the genetic screen. See Materials and methods for further details.

## Data availability

All data generated or analyzed during this study are included in the manuscript and supporting files. Source data files have been provided for Figures 1-6.

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
