## [Editor Report]

The paper focuses on presynaptic homeostatic plasticity (PHP) at the glutamatergic larval *Drosophila* neuromuscular synapse. In this facet of synaptic plasticity, the presynapse increases neurotransmitter release to compensate for diminished postsynaptic sensitivity. To study functional pathways and identify new molecular components of PHP, the authors carried out an electrophysiology-based genetic screen of E3 ubiquitin ligases – key regulators of protein function and degradation pathway and this screen, which forms the backbone of the paper, generated an extensive dataset encompassing 180 genotypes. In follow-up studies, the authors find that the E3 ligase Thin suppresses glutamate release, likely by targeting and downregulating Dysbindin, a transmitter-release-promoting presynaptic protein and based on the experimental data, a model is put forward according to which PHP arises by relieving Dysbindin of Thin-dependent ubiquitination and degradation. This is a strong paper that adds a highly interesting feature to the understanding of the molecular mechanisms that control synaptic strength.

---

## [Decision Letter]

**Decision letter after peer review:**

Thank you for submitting your article "The E3 ligase Thin controls homeostatic plasticity through neurotransmitter release repression" for consideration by *eLife*. Your article has been reviewed by 3 peer reviewers, and the evaluation has been overseen by a Reviewing Editor and Lu Chen as the Senior Editor. The following individuals involved in review of your submission have agreed to reveal their identity: Hiroshi Kawabe (Reviewer #1); Eckart D Gundelfinger (Reviewer #3).

The reviewers have discussed their reviews with one another, and the Reviewing Editor has drafted this to help you prepare a revised submission. There is consensus that your paper is a very strong *eLife* candidate, provided that the reviewers' comments can be addressed. Four items require additional data or experiments (see below).

Recommendations for the Authors

A. Revisions Requiring Additional Experiments

A1. Because global loss of Thin impairs muscle development, the authors re-express Thin solely in the muscle to study synaptic morphology in the absence of presynaptic thin (Figure 5). They find that in this situation presynaptic AZ numbers are significantly increased. The authors attribute this phenotype to unphysiological Thin levels in the muscle rather than to a loss of presynaptic Thin, because postsynaptic Thin overexpression in wild-type animals also increases AZ numbers. This attribution is not fully justifiable without additional data. The precise interpretation of the morphological phenotypes found is important to define the processes that are controlled by Thin. To clarify the issue the authors should analyse NMJ morphology upon presynaptically driven Thin-RNAi as used in the corresponding functional measurements (Figure 4).

A2. Thin was included in the screen in the form of a homozygous mutant allele (∆A) that was previously described to severely affect muscle integrity. As the authors admit, the corresponding reduced miniature amplitudes may confound conclusions – in particular as the QC is increased compared to control. This might imply that the mutant displays some capacity for homeostatic upregulation, at least on a long-term scale. However, Figure 3 shows that such a capacity is not apparent in the absence GluRIIA and that loss of Thin does not further reduce mini amplitudes when GluRIIA is missing. This leaves the reader puzzled without further information, especially on GluRII(A) abundance. It is only with Figure 4 that a motoneuronal RNAi is introduced, importantly with no discernible postsynaptic deficits. It is used to show that an increased RRP underlies increased baseline transmission, whereas SV release probability remains unchanged. The reviewers recommend to introduce this issue earlier (preferentially before Figure 3, which might actually become a supplementary figure) and to further elaborate to address short-term PHP deficits by performing PhTx experiments. Also, the morphological features (mainly Brp, as in Figure 5, maybe in combination with a SV marker) should be explored to increase comparability of the experiments.

A3. Based on the data provided, (partial) co-localization of Dysbindin and Thin at NMJs can only be deduced based on overexpression of fluorescently tagged proteins in motoneurons (Figure 6). This is a substantial shortcoming – but it is probably not easily overcome. Hence, it is important to verify that the two proteins affect each other. In S2 cells, overexpression of Dysbindin in S2 cells leads to striking recruitment of endogenous Thin to the periphery, actually to the very same hot spots (Figure 6). Here, controls are required to rule out that there are any 'bleed-through' effects (here from the green into the magenta channel), and a quantitative image analysis should be carried out. This could, in the best case, make a co-immunoprecipitation dispensable. Beyond this, the authors should explore the possibility of using a similar approach at NMJs instead of in S2 cells, comparing overexpression of tagged versions of Dysbindin alone and Thin alone vs. Dysbindin and Thin co-expressed, e.g. with using Synapsin (or Brp) as localization reference.

A4. It appears very important to test whether Dysbindin expression increases in the absence of Thin, possibly to an extent that endogenous protein becomes directly detectable by an antibody.

B. Revisions Requiring Further Data Analysis, Text Redaction, or Consideration

B1. The mammalian TRIM family is composed of dozens of members. The authors should provide a comparative domain and amino acid sequence analysis to support the notion that Thin is the TRIM32 ortholog. Corresponding sequence analyses could also serve as a basis to discuss related proteins in the fly and possible functional redundancies.

B2. As mentioned in the manuscript and shown by others, presynaptic Dysbindin overexpression increases neurotransmitter release. This is consistent with the model put forward in the current manuscript. Related to increased AZ numbers observed (Figure 5), which the authors do not attribute to loss of presynaptic Thin, the questions arises as to whether presynaptic Dysbindin overexpression increases AZ numbers. If this were not the case, it would support the authors' model, where the Thin-Dysbindin interaction influences synaptic strength at the level of an individual synapse. If the authors already have data related to this issue, they should be included.

B3. The statement that distinct Thin and Dysbindin spots partially overlap is based on nice, high-resolution STED images. However, in terms of quantification the authors only show a single line profile. A more quantitative correlation analysis would strengthen the conclusion and improve the manuscript.

B4. The authors argue that presynaptic Thin remains hardly detectable because of the dominant expression in muscles. The question arises as to whether staining of NMJs in animals with muscle-specific Thin RNAi might overcome this issue and provide information on presynaptic Thin.

B5. The question arises whether there is an observable effect of PhTX treatment on the level of tagged Dysbindin that can be addressed by live imaging.

B6. The cumulative evidence indicates that under baseline conditions, Thin limits SV release via control of Dysbindin (Figure 7). Given that the screen identified quite a few E3 ligases, the question arises as to whether there are candidates in the screen that could be included as example where PHP is affected independently of Dysbindin.

B7. The authors' measure of release probability (Pr) is related to the estimated RRP size, which should be treated with care (see e.g. Neher, Merits and limitations of vesicle pool models in view of heterogeneous populations of synaptic vesicles, Neuron, 2015). For an additional and independent assessment of Pr, the authors should quantify the ratio of 1st and 2nd EPSCs in the 60 Hz trains.

C. Other Issues Requiring Attention

C1. The exact genotypes of all of the *Drosophila* lines used in this study need to be listed in a clear and comprehensive fashion to provide this important information to the reader – analogous to or in Table S1. Ideally, the red colour code of genes in Figure 1E could be used for this new information and for the 'old' Table S1.

C2. Flybase lists 5 publicly available Thin-specific RNAi lines (3 at Bloomington, 2 at VDRC), but the line that was used here seems not to have been specified. This should be rectified. The authors should explain why the specific line was used and whether the use of alternative RNAi lines was considered to validate the observed phenotype and rule out off-target effects.

C3. For the sake of consistency, the genotype thin^ΔA^; 24BGal4>UAS-thin should either be consistently referred to as 'postsynaptic rescue' (Figure 2) or as 'presynaptic thin^ΔA^ mutant' (Figure 5) throughout the manuscript.

C4. The reviewers noticed inconsistencies in the methods section concerning cloning experiments. The mCherry-tagged Thin construct is once referred to as "pUAS_attB_mCherry_thin vector" in the paragraph on plasmid construction, but as "pUAS-thin-mCherry" in the previous paragraph, and as "UAS-thinmcherry" in the Results section. The construct descriptor should be consistent throughout the manuscript. Brief inspection of the primer sequences indicates that mCherry was fused to the C-terminus of Thin, but the restriction sites within the primers do not all match the enzymes that are named as the ones used for cloning. Finally, the authors should indicate which of the Thin splice variants was expressed from the UAS transgene.

C5. The reviewers noted possible inconsistencies in Figure 2. In the top right set of example traces in Figure 2A, the large pink and red traces show a clear difference, which is reflected in the second set of bar diagrams in Figure 2C (i.e. pink and red bars). On the other hand, sky-blue and blue traces on the bottom left in Figure 2A show no difference while the third set of bar diagrams in Figure 2C (i.e. sky blue and blue bars) show a significant difference. This should be explained. In the experiments shown in Figure 4, the authors knocked down Thin in the presynaptic neuron with an intact expression in postsynaptic muscle cells. In Figure 4C, it is shown that this treatment increases EPSCs. This scenario is similar to that in the Thin KO with postsynaptic rescue, shown in Figure 2C (light green bar). However, no difference between gray and light green bars in Figure 2C is apparent. It is unclear how this can be explained.

C6. Figure 5 and Figure S2 indicate striking changes regarding the AZ protein Brp that occur alongside postsynaptic expression of Thin, regardless of WT or mutant background. This might imply an interesting 'retrograde' effects. It might be a good idea to provide a comment on this finding. Further, it seems that at least the Brp labellings in Figure 5 do not at all reflect the different intensities shown in the corresponding graphs. This needs to be explained or other images should be chosen.

C7. The reviewers interpret the images in Figure 6A-6D to originate from mcherry and venus. If so, they should be indicated as such i.e. "Thin-mcherry" and "Dysb-venus" in panels A-D. Otherwise, the reader might assume that endogenous untagged proteins are shown.

C8. There are shortcomings in the documentation of the biochemical part of the cell culture experiments. The Western Blot in Figure 6E is not labelled properly (band at 25 kDa in bottom panel), the bands shown do not really match the data in the bar graph in Figure 6F (e.g. the thin2x sample), and it is unclear what normalization and loading control were used (requires MEMCode or housekeeping protein quantification), and what type of tissue samples from exactly what genotypes were loaded. This should be rectified. Further, there is a faint band at >50 kDa in all lanes of the upper panel of Figure 6E, even in the thin2x lane. This should be discussed. The legend should note that Dysb-Venus was detected with anti-GFP antibodies. The expression level of Thin appears to have been assessed only by the concentration of the transfected DNA rather than by blot analysis or at least determining the actual rate of transfected cells. If true, this should be stated. Further, the methods section mentions that Thin-mCherry was used for transfection and anti-DsRed antibody for detection on blots, but no corresponding results are described.

[Editors' note: further revisions were suggested prior to acceptance, as described below.]

Thank you for resubmitting your work entitled "The E3 ligase Thin controls homeostatic plasticity through neurotransmitter release repression" for further consideration by *eLife*. Your revised article has been evaluated by Lu Chen (Senior Editor) and a Reviewing Editor.

The manuscript has been improved very substantially. No further experiments are required, but there are five remaining issues that need to be addressed by changes to the manuscript, as outlined below:

1. The authors state in their rebuttal that "Under baseline conditions, i.e. in the absence of PhTX, the miniature amplitude was decreased in thinΔA mutants (Figure 2B), after presynaptic and postsynaptic rescue (Figure 2B)". However, in Figure 2B, the gray and light red bars show significant differences, while gray and light blue/light green bars do not. This indicates that mEPSC amplitudes were NOT decreased in thinΔA mutants upon presynaptic or postsynaptic rescue. Can this be clarified or is there an underlying asterisk labeling error? The latter, if relevant, might warrant a final detailed check of the entire manuscript.

2. Demonstrating an increase in Dysbindin levels in thin mutants is important. Many reports have demonstrated reduced expression of alleged "substrates" upon overexpression of a corresponding E3 – as shown in Figure 5. Figure S2C. However, such results have limited value. Overexpression of an E3 can cause artificial ubiquitination by excess enzyme binding with low-affinity to proteins that are not endogenous substrates. Since it is not difficult to detect Dysbindin-Venus by Western blotting, the best experiment under the current circumstances might be to study thin mutant flys expressing low levels of Dysbindin-Venus. At this juncture, the reviewers do not require this experiment to be done. Instead, the authors should conservatively rephrase their conclusion that Dysbindin is regulated by the UPS in a Thin-dependent manner (Figure 6F) and might eliminate "26S" from Figure 6F and change the text accordingly.

3. There appears to be a misunderstanding regarding the reviewers' comment C5 – the concerns regarding discrepancies between Figure 2C and Figure 4C. In the reviewers' view, the expression of thin should be the same in these two scenarios; no thin expression in presynapse but intact expression in postsynapse. There should not be any difference between the results in Figure 2C and Figure 4C. This issue should be explained and resolved.

4. The identity of the transgenic lines shown in Figure 1E should be disclosed. The authors have done a good job of listing all genotypes. However, in its present form, the assignment of individual genotypes to data points on the volcano plot cannot be clearly extracted from the supplementary table. As a compromise, the authors could use the red colour code in the table as previously suggested (comment C1).

5. To avoid word repetition, the reviewers suggest slightly rephrasing the sentence "Regarding the decrease in Brp intensity, Brp intensity was decreased…" (lines 230-231).

---

## [Author Response]

The reviewers have discussed their reviews with one another, and the Reviewing Editor has drafted this to help you prepare a revised submission. There is consensus that your paper is a very strong eLife candidate, provided that the reviewers' comments can be addressed. Four items require additional data or experiments (see below).Recommendations for the AuthorsA. Revisions Requiring Additional ExperimentsA1. Because global loss of Thin impairs muscle development, the authors re-express Thin solely in the muscle to study synaptic morphology in the absence of presynaptic thin (Figure 5). They find that in this situation presynaptic AZ numbers are significantly increased. The authors attribute this phenotype to unphysiological Thin levels in the muscle rather than to a loss of presynaptic Thin, because postsynaptic Thin overexpression in wild-type animals also increases AZ numbers. This attribution is not fully justifiable without additional data. The precise interpretation of the morphological phenotypes found is important to define the processes that are controlled by Thin. To clarify the issue the authors should analyse NMJ morphology upon presynaptically driven Thin-RNAi as used in the corresponding functional measurements (Figure 4).

We fully agree. As suggested, we investigated presynaptic NMJ morphology after presynaptic thin-RNAi expression. In addition, we examined NMJ morphology in homozygous thin^ΔA^ mutants. While we observed a slight increase in Brp number after presynaptic thin-RNAi expression (new Figure 4—figure supplement 1), Brp number and NMJ area were unchanged in thin^ΔA^ mutants (new Figure 3). The observation of largely unchanged NMJ area and Brp number in thin null mutants (new Figure 3), which display a complete PHP block and increased presynaptic release (Figure 2), provides strong genetic evidence that the defects in synaptic physiology are unlikely a consequence of increased NMJ size or AZ number. Conversely, PHP and baseline synaptic transmission were unaffected at NMJs with increased NMJ area and Brp number after postsynaptic thin expression in wild type (new Figure 3—figure supplement 1). Thus, although we cannot exclude that the changes in NMJ size or AZ number seen after postsynaptic thin rescue or presynaptic thin-RNAi expression contribute to the changes in synaptic physiology, our data suggest that the changes in NMJ area and AZ number are separable from changes in synaptic physiology. We updated the Results and Discussion sections correspondingly. => See Results (l. 209), and Discussion (l. 389).

A2. Thin was included in the screen in the form of a homozygous mutant allele (∆A) that was previously described to severely affect muscle integrity. As the authors admit, the corresponding reduced miniature amplitudes may confound conclusions – in particular as the QC is increased compared to control. This might imply that the mutant displays some capacity for homeostatic upregulation, at least on a long-term scale. However, Figure 3 shows that such a capacity is not apparent in the absence GluRIIA and that loss of Thin does not further reduce mini amplitudes when GluRIIA is missing. This leaves the reader puzzled without further information, especially on GluRII(A) abundance. It is only with Figure 4 that a motoneuronal RNAi is introduced, importantly with no discernible postsynaptic deficits. It is used to show that an increased RRP underlies increased baseline transmission, whereas SV release probability remains unchanged. The reviewers recommend to introduce this issue earlier (preferentially before Figure 3, which might actually become a supplementary figure) and to further elaborate to address short-term PHP deficits by performing PhTx experiments. Also, the morphological features (mainly Brp, as in Figure 5, maybe in combination with a SV marker) should be explored to increase comparability of the experiments.

As recommended, we investigated PHP after presynaptic thin-RNAi expression and observed a complete PHP block (Figure 4—figure supplement 1), consistent with the thin^ΔA^ mutant data (Figure 2). Furthermore, we probed Brp and NMJ morphology after presynaptic thin-RNAi expression (see A1, Figure 4—figure supplement 1).

Regarding the reduced miniature amplitude in thin^ΔA^ mutants: Under baseline conditions, i.e. in the absence of PhTX, miniature amplitude was decreased in thin^ΔA^ mutants (Figure 2B), after presynaptic and postsynaptic rescue (Figure 2B), and largely unchanged upon presynaptic thin-RNAi expression (Figure 4B). Quantal content was increased in thin^ΔA^ mutants (Figure 2E), after postsynaptic rescue (Figure 2E), and after presynaptic thin-RNAi expression (Figure 4D). Together, these data imply that the increase in quantal content under baseline conditions induced by presynaptic thin manipulations is separable from a decrease in miniature amplitude.

By definition, PHP is induced by a relative decrease in miniature amplitude. Given that quantal content increased after presynaptic thin manipulations independent of changes in miniature amplitude (see last paragraph), we consider it unlikely that the increased quantal content under baseline conditions represents a homeostatic response.

Miniature amplitude was decreased in thin^ΔA^-GluRIIA double mutants compared to thin^ΔA^ mutants (Figure 2—figure supplement 1B). While a decrease in miniature amplitude induced an increase in quantal content in GluRIIA mutants compared to WT (Figure 2—figure supplement 1D), there was no increase in quantal content in thin^ΔA^-GluRIIA double mutants compared to thin^ΔA^ mutants (Figure 2— figure supplement 1D), implying a PHP defect.

Indeed, miniature amplitude was similar between GluRIIA mutants and thin^ΔA^GluRIIA double mutants (Figure 2—figure supplement 1B). This observation either indicates that we could not resolve a further decrease in miniature amplitude because of signal-to-noise limitations, or that loss of thin reduces GluR levels. Importantly, while miniature amplitudes were similar between GluRIIA mutants and thin^ΔA^-GluRIIA double mutants (Figure 2—figure supplement 1B), EPSC amplitudes were decreased in thin^ΔA^-GluRIIA double mutants, but not in GluRIIA mutants (Figure 2—figure supplement 1C). This suggests a thin-dependent decrease in EPSC amplitude in the GluRIIA mutant background, consistent with impaired PHP.

We summarized and discussed these points in the Discussion (l. 428) and the legend of Figure 2—figure supplement 1.

As suggested, we now show the GluRIIA data as a supplementary figure (Figure 2—figure supplement 1). Based on the fact that we now quantified NMJ morphology in thin^ΔA^ mutants (Figure 3, A1), and that we carried out the formal genetic analysis with presynaptic and postsynaptic rescue in the thin^ΔA^ mutant background (Figure 2), we decided showing the morphology data for thin^ΔA^ mutants and rescue experiments (new Figure 3) before introducing the thin-RNAi physiology and morphology data (Figure 4 and Figure 4—figure supplement 1).

A3. Based on the data provided, (partial) co-localization of Dysbindin and Thin at NMJs can only be deduced based on overexpression of fluorescently tagged proteins in motoneurons (Figure 6). This is a substantial shortcoming – but it is probably not easily overcome. Hence, it is important to verify that the two proteins affect each other. In S2 cells, overexpression of Dysbindin in S2 cells leads to striking recruitment of endogenous Thin to the periphery, actually to the very same hot spots (Figure 6). Here, controls are required to rule out that there are any 'bleed-through' effects (here from the green into the magenta channel), and a quantitative image analysis should be carried out. This could, in the best case, make a co-immunoprecipitation dispensable. Beyond this, the authors should explore the possibility of using a similar approach at NMJs instead of in S2 cells, comparing overexpression of tagged versions of Dysbindin alone and Thin alone vs. Dysbindin and Thin co-expressed, e.g. with using Synapsin (or Brp) as localization reference.

We performed additional experiments and analyses regarding the relationship between Thin and Dysbindin localization in S2 cells and at the NMJ. As recommended, we first investigated a potential bleed-through between the Dysbindin and Thin channel in S2 cells: To test whether the Thin channel (magenta) is excited by the Dysbindin channel (green), we only expressed dysbindin in S2 cells and imaged both channels. Fluorescence intensity did not significantly deviate from background in the Thin channel (magenta) (Figure 5—figure supplement 2A), suggesting no major bleed through from the Dysbindin to the Thin channel. It is also worth noting that both channels were imaged sequentially when expressing both, dysbindin and thin (Figure 5—figure supplement 2A). Finally, although tightly correlated, we also observed a small fraction of uncorrelated Dysbindin and Thin fluorescence (Figure 5—figure supplement 2A). Hence, we consider a major contribution of channel crosstalk to the Dysbindin and Thin localization data in S2 cells unlikely.

Next, we quantified the Pearson’s correlation coefficient (r) for Thin and

Dysbindin fluorescence intensity per pixel in S2 cells and found an average r=0.84

(Figure 5—figure supplement 2B), significantly higher than expected from random Thin and Dysbindin localizations simulated by sampling random point spread function-sized chunks of the data (Figure 5—figure supplement 2B and Material and Methods).

Finally, we studied the relationship between Thin and Dysbindin localization at the NMJ by quantifying the nearest-neighbor distances (NNDs) between Thin and Dysbindin puncta at STED resolution upon overexpression (Figure 5). This analysis revealed significantly smaller NNDs between Thin and Dysbindin puncta compared to simulated random localizations within NMJ boutons (Figure 5). Collectively, these data suggest a relationship between Thin and Dysbindin localization in S2 cells and at the NMJ.

Additionally, we probed Dysbindin localization after overexpression with regard to Synapsin localization at the NMJ of wild type and thin^ΔA^ mutants using STED microscopy. Although we revealed a slight, but significant decrease in NND between Dysbindin and Synapsin puncta in thin^ΔA^ mutants compared to wild type, we also detected a slight increase in Synapsin puncta density in thin^ΔA^ mutants, thereby complicating the interpretation of the results of this experiment. We therefore decided against including these data into the manuscript.

A4. It appears very important to test whether Dysbindin expression increases in the absence of Thin, possibly to an extent that endogenous protein becomes directly detectable by an antibody.

We now performed an anti-dysbindin staining in the thin^ΔA^ mutant background. Unfortunately, we failed to detect a significant anti-dysbindin signal over background fluorescence in thin^ΔA^ mutants. It is worth noting that we quite commonly fail in detecting proteins with low endogenous levels by immunohistochemistry at the *Drosophila* NMJ.

B. Revisions Requiring Further Data Analysis, Text Redaction, or ConsiderationB1. The mammalian TRIM family is composed of dozens of members. The authors should provide a comparative domain and amino acid sequence analysis to support the notion that Thin is the TRIM32 ortholog. Corresponding sequence analyses could also serve as a basis to discuss related proteins in the fly and possible functional redundancies.

We carried out a comparative domain and amino acid sequence analysis between Thin and the human TRIM family (Figure 1—figure supplement 2). Consistent with previous work (LaBeau-DiMenna et al., 2012), this analysis suggests that TRIM32 likely represents the closest human ortholog of thin. In short, out of the large TRIM family, TRIM32 is the only TRIM that has a domain composition that is similar to the one of Thin (N-terminal TRIM followed by C-terminal NHL repeats; Figure 1—figure supplement 2A). Moreover, amino acid sequence alignment of Thin’s RING domains and NHL domains revealed evolutionary conservation of both domains compared to TRIM32, as well as other members of the TRIM C-VII family, which also contains TRIM32 (Ozato et al., 2008; Figure 1—figure supplement 2B). Together, this analysis supports the idea that TRIM32 is Thin’s closest human ortholog.

We updated the manuscript accordingly (l. 142, Figure 1—figure supplement 2).

B2. As mentioned in the manuscript and shown by others, presynaptic Dysbindin overexpression increases neurotransmitter release. This is consistent with the model put forward in the current manuscript. Related to increased AZ numbers observed (Figure 5), which the authors do not attribute to loss of presynaptic Thin, the questions arises as to whether presynaptic Dysbindin overexpression increases AZ numbers. If this were not the case, it would support the authors' model, where the Thin-Dysbindin interaction influences synaptic strength at the level of an individual synapse. If the authors already have data related to this issue, they should be included.

Based on your suggestion, we probed Brp number and NMJ morphology after dysbindin overexpression. This analysis did not yield differences in Brp number or NMJ area compared to controls (Figure 5—figure supplement 1F), thereby further supporting the notion that changes in AZ number/NMJ size are separable from changes in synaptic physiology (see also A1).

B3. The statement that distinct Thin and Dysbindin spots partially overlap is based on nice, high-resolution STED images. However, in terms of quantification the authors only show a single line profile. A more quantitative correlation analysis would strengthen the conclusion and improve the manuscript.

We now quantified the nearest-neighbor distances (NNDs) between Thin and Dysbindin puncta at STED resolution upon overexpression (Figure 5) and observed significantly smaller NNDs between Thin and Dysbindin puncta compared to randomly distributed puncta (Figure 5, see also A3). Note that we decided against conducting a fluorescence intensity-based correlation analysis, because the comparably low fluorescence intensity of STED data has a rather small dynamic range, thus complicating a correlation analysis.

B4. The authors argue that presynaptic Thin remains hardly detectable because of the dominant expression in muscles. The question arises as to whether staining of NMJs in animals with muscle-specific Thin RNAi might overcome this issue and provide information on presynaptic Thin.

Based on this suggestion, we probed endogenous Thin localization in relation to Brp in WT and thin^ΔA^ mutants and observed anti-thin fluorescence in close proximity to Brp in WT, but not in thin^ΔA^ mutants (Figure 5—figure supplement 1D, E). These data suggest that endogenous Thin localizes to presynaptic boutons in addition to postsynaptic muscle cells. We did not repeat these experiments after muscle-specific RNAi expression due to time reasons (l. 292).

B5. The question arises whether there is an observable effect of PhTX treatment on the level of tagged Dysbindin that can be addressed by live imaging.

We conducted this experiment but did not detect obvious differences in anti-GFP fluorescence intensity or localization after venus-dysbindin overexpression between PhTX-treated and untreated NMJs (not shown).

B6. The cumulative evidence indicates that under baseline conditions, Thin limits SV release via control of Dysbindin (Figure 7). Given that the screen identified quite a few E3 ligases, the question arises as to whether there are candidates in the screen that could be included as example where PHP is affected independently of Dysbindin.

Incorporation of other candidate genes identified by the genetic screen would presuppose a rather elaborate set of experiments (genetic rescue, morphology, relationship to dysbindin etc.), similar to the one presented for *thin* in the present study. The realization of these experiments was not feasible during the revision.

B7. The authors' measure of release probability (Pr) is related to the estimated RRP size, which should be treated with care (see e.g. Neher, Merits and limitations of vesicle pool models in view of heterogeneous populations of synaptic vesicles, Neuron, 2015). For an additional and independent assessment of Pr, the authors should quantify the ratio of 1st and 2nd EPSCs in the 60 Hz trains.

We now analyzed the paired-pulse ratio (PPR) of the first two EPSC amplitudes of the 60-Hz train. There was a slight increase in PPR after presynaptic thin-RNAi expression compared to controls (Figure 4H), implying a slight decrease in release probability (pr). By contrast, the train-based pr estimate (first EPSC amplitude/ cum. EPSC amplitude) was similar between thin-RNAi and controls (Figure 4G). Together, these data suggest that the increase in presynaptic release upon presynaptic thinRNAi expression is unlikely caused by an increase in pr, and that presynaptic thinRNAi expression may even lead to a slight pr decrease. We updated the Results section accordingly (l. 263).

C. Other Issues Requiring AttentionC1. The exact genotypes of all of the *Drosophila* lines used in this study need to be listed in a clear and comprehensive fashion to provide this important information to the reader – analogous to or in Table S1. Ideally, the red colour code of genes in Figure 1E could be used for this new information and for the 'old' Table S1.

We updated the manuscript and Table S1 with regard to the exact genotypes (Table S1).

C2. Flybase lists 5 publicly available Thin-specific RNAi lines (3 at Bloomington, 2 at VDRC), but the line that was used here seems not to have been specified. This should be rectified. The authors should explain why the specific line was used and whether the use of alternative RNAi lines was considered to validate the observed phenotype and rule out off-target effects.

We used Bloomington stock 42826 (RRID:BDSC_42826; P{TRiP.HMS02508}attP40). Out of the four stocks available from Bloomington, three express dsRNA for RNAi of thin under UAS control in the pVALIUM20 vector, one of the second-generation TRiP knockdown vectors (Ni et al., 2011; Perkins et al., 2015). The other line available from Bloomington was made with the older, firstgeneration pVALIUM1 vector. Out of the three VALIUM20 stocks available, we used stock 42826, because it is the only one that was previously published in the context of genetic screens (Kuleesha et al., 2016, Rotelli et al., 2019). We neither ordered nor tested additional thin RNAi lines from Bloomington or VDRC because of time reasons. We updated the methods section correspondingly (Key resources table, an l. 482).

C3. For the sake of consistency, the genotype thin ^ΔA^; 24BGal4>UAS-thin should either be consistently referred to as 'postsynaptic rescue' (Figure 2) or as 'presynaptic thin ^ΔA^ mutant' (Figure 5) throughout the manuscript.

We now consistently use the term ‘postsynaptic rescue’ throughout the manuscript.

C4. The reviewers noticed inconsistencies in the methods section concerning cloning experiments. The mCherry-tagged Thin construct is once referred to as "pUAS_attB_mCherry_thin vector" in the paragraph on plasmid construction, but as "pUAS-thin-mCherry" in the previous paragraph, and as "UAS-thinmcherry" in the Results section. The construct descriptor should be consistent throughout the manuscript. Brief inspection of the primer sequences indicates that mCherry was fused to the C-terminus of Thin, but the restriction sites within the primers do not all match the enzymes that are named as the ones used for cloning. Finally, the authors should indicate which of the Thin splice variants was expressed from the UAS transgene.

We are sorry about these inconsistencies and now consistently labelled the construct as “pUAS-mCherry-thin”, “Thin^mCherry^”. Moreover, we added the correct primer sequences and the Ref.Seq number for the expressed Thin isoform (isoform A) to the methods section (Material and Methods, l. 509).

Note that we now performed additional cloning experiments for the new Western blot analysis (see C8).

C5. The reviewers noted possible inconsistencies in Figure 2. In the top right set of example traces in Figure 2A, the large pink and red traces show a clear difference, which is reflected in the second set of bar diagrams in Figure 2C (i.e. pink and red bars). On the other hand, sky-blue and blue traces on the bottom left in Figure 2A show no difference while the third set of bar diagrams in Figure 2C (i.e. sky blue and blue bars) show a significant difference. This should be explained. In the experiments shown in Figure 4, the authors knocked down Thin in the presynaptic neuron with an intact expression in postsynaptic muscle cells. In Figure 4C, it is shown that this treatment increases EPSCs. This scenario is similar to that in the Thin KO with postsynaptic rescue, shown in Figure 2C (light green bar). However, no difference between gray and light green bars in Figure 2C is apparent. It is unclear how this can be explained.

Regarding the representative presynaptic thin rescue data (sky blue and dark blue, Figure 2): We apologize for having chosen a representative cell for the presynaptic rescue group with PhTX (dark blue) that did not reflect the group average and now updated the example trace correspondingly (dark blue traces in Figure 2A). Most likely, the difference in EPSC amplitude between PhTX-treated and untreated cells after presynaptic thin rescue is a result of incomplete rescue, a phenomenon quite frequently observed after overexpression-based genetic rescue at the *Drosophila* NMJ. Alternatively, the smaller EPSC amplitudes after rescue may arise from defects in muscle architecture (LaBeau-DiMenna et al., 2012). Note, however, that there was a significant increase in quantal content after presynaptic rescue (Figure 2D), suggesting a partial PHP rescue. We now discuss this in the updated Results section (l. 160).

Concerning the comparison of EPSC amplitudes between thin^ΔA^ presynaptic rescue (Figure 2C) and presynaptic thin-RNAi (Figure 4C): If we understand the concern correctly, then the question is why the increase in EPSC amplitude after presynaptic thin-RNAi expression (Figure 4C) is more pronounced than the one after postsynaptic rescue (Figure 2C, light green data). The short answer is that this is most likely due to different effects on mEPSC amplitude in these genetic backgrounds: mEPSC amplitudes were decreased in thin^ΔA^ (Figure 2B, light red) and – to a lesser extend – after postsynaptic rescue (Figure 2B, light green), but largely unchanged after presynaptic thin-RNAi expression (Figure 4B). In combination with largely unchanged EPSC amplitudes in thin^ΔA^ (Figure 2C, light red), slightly increased EPSC amplitudes after postsynaptic rescue (Figure 2C, light green), and increased EPSP amplitudes after presynaptic thin-RNAi expression (Figure 4C), this translates into a similar increase in quantal content in these three genotypes (Figure 2E and 4D). Hence, thin perturbation results in enhanced neurotransmitter release under baseline conditions in three independent experiments.

We discuss the relationship between miniature amplitude and presynaptic release in the Discussion (l. 428).

C6. Figure 5 and Figure S2 indicate striking changes regarding the AZ protein Brp that occur alongside postsynaptic expression of Thin, regardless of WT or mutant background. This might imply an interesting 'retrograde' effects. It might be a good idea to provide a comment on this finding. Further, it seems that at least the Brp labellings in Figure 5 do not at all reflect the different intensities shown in the corresponding graphs. This needs to be explained or other images should be chosen.

Indeed, Brp intensity was decreased in thin^ΔA^ mutants, after presynaptic rescue (Figure 3E), and postsynaptic thin overexpression in wild type (new Figure 3—figure supplement 1E). We also observed a trend towards decreased Brp intensity after postsynaptic rescue (Figure 3). However, while Brp intensity was decreased after presynaptic rescue (new Figure 3) and postsynaptic thin overexpression (new Figure 3—figure supplement 1), PHP and baseline synaptic transmission were unchanged in both genotypes (Figure 2, new Figure 3—figure supplement 1). Conversely, Brp levels were unaffected by presynaptic thin-RNAi expression, but PHP was impaired, and baseline synaptic transmission increased (Figure 4, new Figure 4—figure supplement 1). Together, these results again support the idea that changes in NMJ morphology are separable from changes in synaptic physiology after genetic thin manipulations (see A1). Although possible, we therefore consider it unlikely that the decrease in Brp intensity is a major factor underlying the PHP defect or the increase in synaptic transmission after presynaptic thin perturbation.

We now discuss the changes in Brp intensity in the Results (l. 209) and Discussion sections (l. 389).

Finally, we updated the example image in the new version of Figure 3 to better reflect the average Brp data.

C7. The reviewers interpret the images in Figure 6A-6D to originate from mcherry and venus. If so, they should be indicated as such i.e. "Thin-mcherry" and "Dysb-venus" in panels A-D. Otherwise, the reader might assume that endogenous untagged proteins are shown.

We apologize for not specifying the tag in the images. As specified in the figure legend, the images indeed show pUAS-mCherry-thin (‘Thin^mCherry^’) and pUAS-venusDysbindin (‘Dysbindin^venus^’). We updated the new Figure 5 correspondingly.

C8. There are shortcomings in the documentation of the biochemical part of the cell culture experiments. The Western Blot in Figure 6E is not labelled properly (band at 25 kDa in bottom panel), the bands shown do not really match the data in the bar graph in Figure 6F (e.g. the thin2x sample), and it is unclear what normalization and loading control were used (requires MEMCode or housekeeping protein quantification), and what type of tissue samples from exactly what genotypes were loaded. This should be rectified. Further, there is a faint band at >50 kDa in all lanes of the upper panel of Figure 6E, even in the thin2x lane. This should be discussed. The legend should note that Dysb-Venus was detected with anti-GFP antibodies. The expression level of Thin appears to have been assessed only by the concentration of the transfected DNA rather than by blot analysis or at least determining the actual rate of transfected cells. If true, this should be stated. Further, the methods section mentions that Thin-mCherry was used for transfection and anti-DsRed antibody for detection on blots, but no corresponding results are described.

We are sorry about the shortcomings regarding the documentation of the biochemistry and cell culture experiments. Based on this concern, we repeated the Western blot analysis. Since it was difficult to detect mCherry-tagged Thin, we generated and expressed HA-tagged Thin (pUAS-HA-thin). However, it was also challenging to detect pUAS-HA-thin on Western blots (Figure 5—figure supplement 2C). This may be due to Thin’s large disordered domains and/or its comparably large size (Figure 1—figure supplement 2A). Quantification of the Thin/Tubulin ratio at 1x and 2x thin DNA concentration revealed an increase of Thin/Tubulin that correlated with HA-thin DNA concentration (not shown). However, the low signal-to-noise ratio of the Thin signal precludes a meaningful quantification and interpretation of these data. We thus plotted Dysbindin/Tubulin for cells transfected without, 1x and 2x pUAS-HA-thin (now specified in the Methods section). Note that we are confident about a robust thin transfection efficiency, because we consistently observed mCherry-positive cells in the old experiments, as well as anti-HA signal in the last rounds of Western blots.

We chose a new example that better reflects the average data (Figure 5— figure supplement 2C). Note that – besides the fact that most blots showed some ‘cosmetic flaws’ (Figure 5—figure supplement 2C); we explicitly show a representative instead of the best example – we consistently observed a reduction in anti-GFP/anti-Tubulin that correlated with pUAS-HA-thin concentration, thereby strongly supporting the idea that Thin degrades Dysbindin in *Drosophila*.

The new blot does not display faint bands at >60kDa. We also verified that the new blot is labelled correctly, including the correct molecular sizes.

The legend now specifies that pUAS-venus-Dysbindin (‘Dysbindin^venus^’) and pUAS-HA-thin (‘Thin^HA^’) were detected with anti-GFP and anti-HA, respectively. The quantification of anti-GFP (Dysbindin^venus^) was now normalized to anti-Tubulin (Figure 5—figure supplement 2D). The Methods section was updated correspondingly.

[Editors' note: further revisions were suggested prior to acceptance, as described below.]

The manuscript has been improved very substantially. No further experiments are required, but there are five remaining issues that need to be addressed by changes to the manuscript, as outlined below:1. The authors state in their rebuttal that "Under baseline conditions, i.e. in the absence of PhTX, the miniature amplitude was decreased in thinΔA mutants (Figure 2B), after presynaptic and postsynaptic rescue (Figure 2B)". However, in Figure 2B, the gray and light red bars show significant differences, while gray and light blue/light green bars do not. This indicates that mEPSC amplitudes were NOT decreased in thinΔA mutants upon presynaptic or postsynaptic rescue. Can this be clarified or is there an underlying asterisk labeling error? The latter, if relevant, might warrant a final detailed check of the entire manuscript.

mEPSC amplitudes were decreased after presynaptic rescue (p<0.001) and

postsynaptic rescue (p=0.04) in the thinΔA mutant background compared to WT. Hence, there was indeed an asterisk labeling error in Figure 2B. We are sorry about this mistake and thank the reviewers for catching it. We now updated Figure 2B correspondingly.

2. Demonstrating an increase in Dysbindin levels in thin mutants is important. Many reports have demonstrated reduced expression of alleged "substrates" upon overexpression of a corresponding E3 – as shown in Figure 5. Figure S2C. However, such results have limited value. Overexpression of an E3 can cause artificial ubiquitination by excess enzyme binding with low-affinity to proteins that are not endogenous substrates. Since it is not difficult to detect Dysbindin-Venus by Western blotting, the best experiment under the current circumstances might be to study thin mutant flys expressing low levels of Dysbindin-Venus. At this juncture, the reviewers do not require this experiment to be done. Instead, the authors should conservatively rephrase their conclusion that Dysbindin is regulated by the UPS in a Thin-dependent manner (Figure 6F) and might eliminate "26S" from Figure 6F and change the text accordingly.

We completely agree and rephrased our conclusions regarding Dysbindin regulation by the UPS throughout the text. We also directly address this point in the Results section:

"Although we cannot exclude the possibility that Thin overexpression induced artificial Dysbindin ubiquitination by excess enzyme binding with low affinity, these data are consistent with the idea that Thin acts as an E3 ligase for Dysbindin in *Drosophila*, similar to TRIM32 in humans (Locke et al., 2009)." (l. 330). Moreover, we eliminated the model shown in Figure 6F to avoid any confusion.

3. There appears to be a misunderstanding regarding the reviewers' comment C5 – the concerns regarding discrepancies between Figure 2C and Figure 4C. In the reviewers' view, the expression of thin should be the same in these two scenarios; no thin expression in presynapse but intact expression in postsynapse. There should not be any difference between the results in Figure 2C and Figure 4C. This issue should be explained and resolved.

We agree – in theory, postsynaptic thin rescue (Figure 2C) should produce a similar phenotype as presynaptic thin-RNAi expression (Figure 4C). Indeed, the increase in quantal content under baseline conditions is very similar between postsynaptic thin rescue (Figure 2E, light green bar) and presynaptic thin-RNAi expression (Figure 4D), in line with a model in which presynaptic thin perturbation increases quantal content in both genotypes. The smaller mEPSC and EPSC amplitudes under baseline conditions after postsynaptic thin rescue (Figure 2B, C) compared to thin-RNAi (Figure 4B, C) are most likely due to non-endogenous Thin levels caused by thin overexpression in the thinΔA mutant background. We now discuss this possibility in the main text: "Note that the smaller mEPSC and EPSC amplitudes under baseline conditions after postsynaptic thin rescue (Figure 2B, C) compared to thin^RNAi^ (Figure 4B, C) are most likely due to non-endogenous postsynaptic Thin levels caused by thin overexpression in the thin^ΔA^ mutant background." (l. 256)

4. The identity of the transgenic lines shown in Figure 1E should be disclosed. The authors have done a good job of listing all genotypes. However, in its present form, the assignment of individual genotypes to data points on the volcano plot cannot be clearly extracted from the supplementary table. As a compromise, the authors could use the red colour code in the table as previously suggested (comment C1).

As suggested, we now labeled transgenic lines with significantly altered EPSC amplitude in red in the supplementary table and sorted the lines according to EPSC amplitude.

5. To avoid word repetition, the reviewers suggest slightly rephrasing the sentence "Regarding the decrease in Brp intensity, Brp intensity was decreased…" (lines 230-231).

We changed the text correspondingly: "Furthermore, Brp intensity was decreased after presynaptic rescue (…)." (l. 225).